



# Simulation of snow albedo and solar irradiance profile with the two-stream radiative transfer in snow (TARTES) v2.0 model

Ghislain Picard[1] and Quentin Libois[2]

[1]Univ. Grenoble Alpes, CNRS, IGE, 38000 Grenoble, France
[2]CNRM, Université de Toulouse, Météo-France, CNRS, Toulouse, France

**Correspondence:** Ghislain Picard (ghislain.picard@univ-grenoble-alpes.fr)

**Abstract.** The Two-streAm Radiative TransfEr in Snow (TARTES) model computes the spectral albedo and the profiles of spectral absorption, irradiance and actinic fluxes for a multi-layer plane-parallel snowpack. Each snow layer is characterized by its specific surface area, density, and impurities content, in addition to shape parameters. In the landscape of snow optical numerical models, TARTES distinguishes itself by taking into account different shapes of the particles through two shape parameters, namely the absorption enhancement parameter $B$ and the asymmetry factor $g$. This is of primary importance as recent studies working at the microstructure level have demonstrated that snow does not behave as a collection of equivalent ice spheres, a representation widely used in other models. Instead, $B$ and $g$ take specific values that do not correspond to any simple geometrical shape, which leads to the concept of "optical shape of snow". Apart from this specificity, TARTES combines well established radiative transfer principles to compute the scattering and absorption coefficients of pure or polluted snow, and the $\delta$-Eddington two-stream approximation to solve the multi-layer radiative transfer equation. The model is implemented in Python, but conducting TARTES simulations is also possible without any programming through the SnowTARTES web application, making it very accessible to non-experts and for teaching purposes. Here, after describing the theoretical and technical details of the model, we illustrate its main capabilities and present some comparisons with other common snow radiative transfer models (AART, DISORT-Mie, SNICAR-ADv3) as a validation procedure. Overall the agreement on the spectral albedo, when in compatible conditions (i.e. with spheres), is usually within 0.02, and is better in the visible and near-infrared compared to longer wavelengths of the solar domain.

## 1 Introduction

Snow, a porous medium made of ice and air, is by far the most reflective material in the solar spectral domain on Earth. Any fluctuation of the snow cover extent or changes in the surface snow properties have consequences on the global radiative budget, and on the climate (Qu and Hall, 2007; Räisänen et al., 2017). Snow albedo (also known as hemispherical reflectance, Schaepman-Strub et al., 2006) is the primary variable controlling the amount of solar energy absorbed in the snowpack (Flanner et al., 2011). Also of importance is the depth at which this absorption occurs. The deeper solar radiation is absorbed, the less likely the corresponding heat is to be transferred back to the surface by thermal conduction where it can eventually be evacuated to the atmosphere through longwave emission or turbulent mixing. Hence, the warming and potential melt of the snowpack



depend on the vertical profile of absorbed sunlight (Dombrovsky et al., 2019). This profile is often approximated by an exponential function decreasing with depth, with the decay length called the $e$-folding depth or penetration depth (Kokhanovsky, 2022). This quantity is also of great importance for photochemical processes (King and Simpson, 2001; Domine et al., 2008).

   Snow optical properties are driven by the physical properties of the snow and the impurities it contains, along with the illumination conditions. The snow microstructure (i.e. the arrangement of ice and air at the micrometer scale) controls the
absorption of radiation in the near-infrared (Wiscombe and Warren, 1980), and is key to understand some of the snow-albedo feedback loops that amplify climate change in snow-covered regions (Hall, 2004; Qu and Hall, 2007; Picard et al., 2012; Box et al., 2022). It is common to represent snow microstructure as a collection of grains with prescribed shape and size (Warren and Wiscombe, 1980; Grenfell and Warren, 1999). However, it is known that snow on the ground is generally not granular, so that equivalent concepts have emerged to quantify more general microstructures. For instance the specific surface area (SSA)
(the ratio between the air-interface surface area and the mass of ice, Domine et al., 2006) advantageously replaces the grain size as it can be rigorously defined and calculated for any porous medium made of two phases (ice and air here). Regarding the shape, the situation is less advanced but geometrical metrics related to the chord length distribution can provide useful information (Malinka, 2014; Krol and Löwe, 2016; Dumont et al., 2021; Robledano et al., 2023).

   The thickness of the snow cover is another major driving variable in the case of shallow covers and a dark underlying ground,
especially in the visible where light penetrates deepest (Perovich, 2007). Another important variable is the roughness of the surface which tends to decrease the albedo (Warren et al., 1998; Leroux and Fily, 1998; Larue et al., 2020). The presence of liquid water also slightly changes the absorption coefficient in a few spectral bands (e.g. 980–1000 nm), which can be detected in spectral albedo measurements (Dumont et al., 2017; Donahue et al., 2022). More importantly it induces fast structural changes of the microstructure (Colbeck, 1982; Brun et al., 1989), usually leading to a rapid decrease in SSA and thus in
albedo . Impurities, whether they are of mineral, organic, or biological origin, can also greatly affect the absorption in the visible and drive both the albedo and the penetration (Chevrollier et al., 2022; Réveillet et al., 2022; Di Mauro et al., 2024). At last, the illumination characteristics (angular and spectral distributions) also play a role because the snow reflectance depends on the wavelength and the incidence angle. As such, neither broadband nor spectral snow albedo are strict surface snow properties, because of this dependency on the illumination characteristics. It means than even without any change of the snow
properties, the surface albedo and penetration depth may change with changing environmental conditions (presence of clouds, sun elevation, etc.).

   To account for these multiple factors numerous snow optical models have been developed. Even though most rely on the radiative transfer (RT) principles, they greatly differ in the method used to solve the RT equation (RTE, e.g. Chandrasekhar, 1960), in the representation of the medium (snow microstructure, the 1D or 3D geometry of the snowpack, surface roughness,
heteorogeneity of the snowpack, impurities, etc.), in the fundamental constants used (e.g. real and imaginary parts of the ice refractive index) and in the output optical quantities (e.g. albedo, absorption profile, actinic flux, transmittance). Each model has its niche of applications, from efficient but approximate code suitable to large scale climate models, to very precise solvers to investigate detailed optical behaviors (e.g. bi-directional distribution reflectance function, BRDF). To cite a few, pioneering work used phenomenological (Dunkle and Bevans, 1956; Bohren, 1987) or more rigorous (Wiscombe and Warren, 1980;



Warren and Wiscombe, 1980) two-stream approximations to solve the RTE, and Mie theory (Mie, 1908) or geometrical optics
(Bohren and Barkstrom, 1974) to represent spherical ice particles suspended in the air. DISORT (Stamnes et al., 1988a, b) is
a general robust and popular solver frequently applied to snow, usually in combination with the Mie theory, hence assuming
spherical grains (Glendinning and Morris, 1999; Green et al., 2002; Gallet et al., 2011; Carmagnola et al., 2013; Dang et al.,
2019). Here, we refer to this combination as "DISORT-Mie". An advantage of DISORT is its ability to account for the radiation

propagation in many directions (multi-stream). To account for other particle shapes and calculate the BRDF, an efficient code
was proposed by Mishchenko et al. (1999) (available as a python package here: https://github.com/ghislainp/mishchenko_brf,
last access: 24 March 2024) but it cannot handle layering. TUV-Snow is a DISORT-based coupled snow-atmosphere model
specifically designed for UV radiation and photochemistry applications (Lee-Taylor and Madronich, 2002; France et al., 2011).
PBSAM (Aoki et al., 2011) and SNICAR (Flanner and Zender, 2005) are fast two-stream solvers suitable for surface albedo

calculation in climate simulations (Onuma et al., 2020; Usha et al., 2020). SNICAR is currently one of the most actively
developed code with a large panel of state-of-the-art parameterizations, for instance to account for snow algae (Cook et al.,
2017) or to account for ice layers with the recent replacement of the two-stream by the adding-doubling solver (Flanner et al.,
2021). Importantly, all these models are uni-dimensional, meaning that they describe the snowpack as a stack of homogeneous,
horizontally infinite and flat layers. This configuration is known as plane-parallel. Conversely three-dimensional RT models

are necessary to account for snowpack with 3D structures, embedded objects or light sources. Some examples are models for
rough surfaces (Warren et al., 1998; Larue et al., 2020; Robledano et al., 2022), for explicit photon trajectory calculation in the
snow microstructure (Kaempfer et al., 2005; Picard et al., 2009; Xiong and Shi, 2014; Letcher et al., 2022; Robledano et al.,
2023) and for interaction with embedded objects or instruments (Gallet et al., 2009; Picard et al., 2016).

In this rich landscape, the Two-streAm Radiative TransfEr in Snow (TARTES) spectral model differs by two main aspects.

First it relies on a simple yet state-of-the-art representation of the snow microstructure, allowing to represent it with four pa-
rameters only: the SSA, the density, the asymmetry factor $g$ that quantifies the forward scattering of snow and the absorption
enhancement parameter $B$ that quantifies the lengthening of photons paths inside the ice phase due to multiple internal reflec-
tions. As a consequence TARTES is not restricted to spherical particles and any couple of $g$ and $B$ values can be used, possibly
not corresponding to any particular idealized geometrical shape. It also offers the possibility to conform with the Asymptotic

Approximation Radiative Transfer (AART Kokhanovsky and Zege, 2004) when the snowpack is semi-infinite, while still being
able to simulate a multi-layered snowpack. The second main difference from other models is the use of the Python language.
This facilitates rapid tests (e.g. in notebooks) and implementation of new features (e.g. impurities, Tuzet et al., 2019). While
TARTES has been used over a decade (e.g. Libois et al., 2013; Shao et al., 2018; van Dalum et al., 2019; Tuzet et al., 2020;
Manninen et al., 2021; Veillon et al., 2021), this paper aims at a first comprehensive and formal description of the model. Some

minor but significant adjustments using the latest results on microstructure (Robledano et al., 2023) are also included, as well
as a brief presentation of the ecosystem of tools relevant to snow optical computations built around TARTES.

Section 2 provides the detailed derivation of the model, Sect. 3 presents the TARTES software and the associated ecosystem,
and Sect. 4 presents the results including self consistency checks, a comparison with other models, and a presentation of its




specific capabilities. Section 5 discusses how TARTES fits with the concept "optical shape of the snow" and presents the model
limitations. Section 6 concludes this study.

## 2   The physics behind TARTES

TARTES relies on the $\delta$-Eddington approximation to solve the plane-parallel RTE and compute the spectral upwelling and
downwelling fluxes within a multi-layered snowpack. To this end the single scattering properties of each layer are computed
from the SSA, density, snow grain shape and amount of light absorbing impurities (Fig. 1). This section provides all the
theoretical details on which TARTES is built, and new formulations added in the version v2.0.

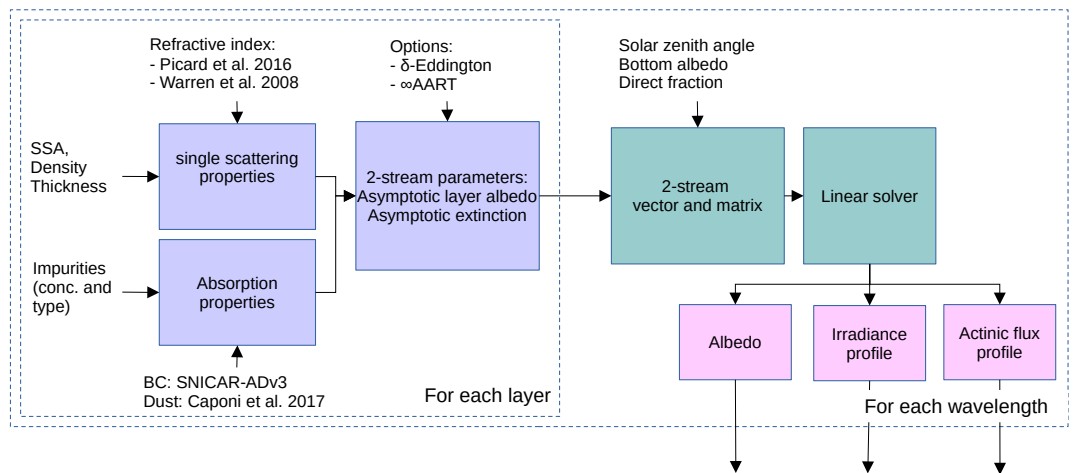

**Figure 1.** Main inputs, options and flow of computations in TARTES v2.0.

### 2.1   The plane-parallel radiative transfer equation

The steady unpolarized RTE describes the intensity (or radiance) field $I$ propagating in an absorbing and scattering slab at
depth $z$ in a direction $(\theta, \phi)$, where $\phi$ is the azimuth angle, $\theta$ the zenith angle is defined as the angle between the inward normal
to the surface and the direction of light propagation, and $z$ is defined positive from the surface downward. Such a medium is
characterized by its extinction coefficient $\sigma_e$ (m$^{-1}$), scattering coefficient $\sigma_s$ (m$^{-1}$) and scattering phase function $p(\theta, \phi, \theta', \phi')$
(unitless). The absorption coefficient is obtained by energy conservation as $\sigma_a = \sigma_e - \sigma_s$. The phase function describes the
probability for light to be scattered into the direction $(\theta, \phi)$ when coming from the direction $(\theta', \phi')$. Here we consider an
horizontal multi-layered snowpack. Each layer of the snowpack is assumed to be isotropic and to have homogeneous optical
properties. The snowpack is illuminated at the surface by solar radiation, that can be a combination of diffuse (i.e. isotropic)
and direct light incident at zenith angle $\theta_0$ and azimuth angle $\phi_0$. There are no internal light sources in the snowpack and no
thermal emission is considered since TARTES focuses on the shortwave range $(200 - 4000\,\text{nm})$. Along the direction $s$ defined
by $(\theta, \phi)$, $I$ decreases due to extinction (absorption and scattering) and increases due to scattering from all other directions





$(\theta', \phi')$, so that the RTE reads:

$$\frac{dI(z,\theta,\phi)}{ds} = -\sigma_e I(z,\theta,\phi) + \frac{\sigma_s}{4\pi} \int_0^\pi \int_0^{2\pi} p(\theta,\phi,\theta',\phi')I(z,\theta',\phi')d\phi'\sin\theta'd\theta', \tag{1}$$

where the phase function is normalized so that $\frac{1}{4\pi}\int_0^\pi \int_0^{2\pi} p(\theta,\phi,\theta',\phi')d\phi'\sin\theta'd\theta' = 1$. Defining $\mu = \cos\theta$ and $\mu' = \cos\theta'$,

noting that $dz = \mu ds$, and further defining the optical thickness such that $d\tau = \sigma_e dz$, Eq. 1 becomes:

$$\mu\frac{dI(\tau,\mu,\phi)}{d\tau} = -I(\tau,\mu,\phi) + \frac{\omega}{4\pi}\int_{-1}^1 \int_0^{2\pi} p(\mu,\phi,\mu',\phi')I(\tau,\mu',\phi')d\phi'd\mu', \tag{2}$$

where the single scattering albedo $\omega = \sigma_s/\sigma_e$. Since TARTES focuses on radiative fluxes through horizontal surfaces, Eq. 2 can be azimuthally integrated, which reads:

$$\mu\frac{dI(\tau,\mu)}{d\tau} = -I(\tau,\mu) + \frac{\omega}{2}\int_{-1}^1 p(\mu,\mu')I(\tau,\mu')d\mu', \tag{3}$$

where we have defined the azimuthally-averaged intensity $I(\tau,\mu) = \frac{1}{2\pi}\int_0^{2\pi} I(\tau,\mu,\phi)d\phi$, and the azimuth-independent phase

function $p(\mu,\mu') = \frac{1}{2\pi}\int_0^{2\pi} p(\mu,\phi,\mu',\phi')d\phi$.

When the snowpack is illuminated by a beam source (e.g. direct solar radiation) it is useful to write $I(\tau,\mu) = I_{\text{dir}}(\tau,\mu) + I_{\text{diff}}(\tau,\mu)$, where the direct intensity $I_{\text{dir}}$ corresponds to light that has not been scattered and $I_{\text{diff}}$ is the diffuse intensity. At the

surface the direct intensity is $F_0\delta(\mu-\mu_0)\delta(\phi-\phi_0)$, where $\mu_0 = \cos\theta_0$ and at depth:

$$I_{\text{dir}}(\tau,\mu) = \frac{F_0}{2\pi}\delta(\mu-\mu_0)e^{-\tau/\mu_0}. \tag{4}$$

Reporting Eq. 4 in Eq. 3, we obtain the RTE for the diffuse intensity:

$$\mu\frac{dI_{\text{diff}}(\tau,\mu)}{d\tau} = -I_{\text{diff}}(\tau,\mu) + \frac{\omega}{2}\int_{-1}^1 p(\mu,\mu')I_{\text{diff}}(\tau,\mu')d\mu' + \frac{\omega}{4\pi}p(\mu,\mu_0)F_0 e^{-\tau/\mu_0}. \tag{5}$$

From now on $I_{\text{diff}}$ will simply be referred as $I$.

### 2.2   The $\delta$-Eddington approximation of the phase function


Within each layer snow is assumed isotropic, so that the phase function depends only on the scattering angle $\Theta$ between the incident and scattered light and we can write $p(\mu,\phi,\mu',\phi') = p(\cos\Theta)$. This angle is such that:

$$\cos\Theta = \mu\mu' + \sqrt{(1-\mu^2)(1-\mu'^2)}\cos(\phi-\phi'). \tag{6}$$





$p(\cos\Theta)$ can be expanded in Legendre polynomials $P_l$:

$$p(\cos\Theta) = \sum_{l=0}^{\infty} \omega_l P_l(\cos\Theta), \quad \text{where } \omega_l = \frac{2l+1}{2} \int_{-1}^{1} p(\cos\Theta) P_l(\cos\Theta) \, \mathrm{d}\cos\Theta. \tag{7}$$

By virtue of normalization $\omega_0 = 1$, and the mean cosine of the scattering angle of the phase function, called the asymmetry factor $g$, is such that $\omega_1 = 3g$. Using the addition theorem of spherical harmonics (Chandrasekhar, 1960) it can finally be shown that:

$$p(\mu, \mu') = \sum_{l=0}^{\infty} \omega_l P_l(\mu) P_l(\mu'). \tag{8}$$

Hence the two-term truncation of the phase function reads:

$$p(\mu, \mu') = 1 + 3g\mu\mu'. \tag{9}$$

To handle the strong forward scattering of snow particles, TARTES relies on the $\delta$-Eddington approximation, which consists in writing the phase function as the sum of a strictly forward scattering component (a Dirac) and a two-term phase function (Eq. 9). Joseph et al. (1976) proposed to weight both contributions so that the asymmetry factor is conserved, and the second

moment of the phase function equals $g^2$ (i.e. the second moment of the Henyey-Greenstein phase function (Henyey and Greenstein, 1941) with asymmetry factor $g$). This reads:

$$p(\mu, \mu') = 2g^2 \delta(\mu - \mu') + (1 - g^2)(1 + 3g^*\mu\mu') \tag{10}$$

with $g^* = \dfrac{g}{1+g}$. Combining Eqs 5 and 10 we obtain:

$$\mu \frac{\mathrm{d}I(\tau^*, \mu)}{\mathrm{d}\tau} = -I(\tau^*, \mu) + \frac{\omega^*}{2} \int_{-1}^{1} (1 + 3g^*\mu\mu') I(\tau^*, \mu') \mathrm{d}\mu' + \frac{\omega^*}{4\pi}(1 + 3g^*\mu\mu_0) F_0 e^{-\tau^*/\mu_0}, \tag{11}$$

where the following variable changes have been made:

$$\tau^* = \tau(1 - \omega g^2), \tag{12}$$

$$\omega^* = \frac{(1 - g^2)\omega}{(1 - \omega g^2)}. \tag{13}$$

Hence the $\delta$-Eddington approximation of the phase function consists in solving Eq. 5, with $\tau$, $\omega$ and $g$ replaced by $\tau^*$, $\omega^*$ and $g^*$. $g^*$ is less than $g$ so that the scaled phase function is less forward-peaking than the original phase function, which will reduce

errors in the following two-stream resolution of the RTE. Note that in this approximation the solution for direct radiation is scaled accordingly, meaning that direct radiation can propagate deeper in the snowpack, because light scattered in the forward direction is treated as unscattered light.



### 2.3 Equations for fluxes and Eddington approximation

In TARTES, we are interested in the vertical downward and upward fluxes in the snowpack, $F^-$ and $F^+$ respectively. These quantities are defined as:

$$F^-(\tau^*) = 2\pi \int_0^1 I(\tau^*, \mu)\mu \mathrm{d}\mu, \tag{14}$$

$$F^+(\tau^*) = 2\pi \int_0^1 I(\tau^*, -\mu)\mu \mathrm{d}\mu. \tag{15}$$

Integrating Eq. 11 over both positive and negative values of $\mu$ results into two differential equations:

$$\frac{dF^-(\tau^*)}{d\tau^*} = -2\pi \int_0^1 I(\tau^*, \mu)\mathrm{d}\mu + \pi\omega^* \int_0^1 \int_{-1}^1 (1 + 3g^*\mu\mu')I(\tau^*, \mu')\mathrm{d}\mu'\mathrm{d}\mu + \omega^*\gamma_4 F_0 e^{-\tau^*/\mu_0}, \tag{16}$$

$$\frac{dF^+(\tau^*)}{d\tau^*} = 2\pi \int_0^1 I(\tau^*, -\mu)\mathrm{d}\mu - \pi\omega^* \int_0^1 \int_{-1}^1 (1 - 3g^*\mu\mu')I(\tau^*, \mu')\mathrm{d}\mu'\mathrm{d}\mu - \omega^*\gamma_3 F_0 e^{-\tau^*/\mu_0}, \tag{17}$$

with

$$\gamma_4 = \frac{1}{4}(2 + 3g^*\mu_0) \quad \text{and} \quad \gamma_3 = \frac{1}{4}(2 - 3g^*\mu_0). \tag{18}$$

Next the Eddington approximation is used, which consists in expanding the intensity $I(\tau^*, \mu)$ as:

$$I(\tau^*, \mu) = I_0(\tau^*) + \mu I_1(\tau^*), \tag{19}$$

so that:

$$F^-(\tau^*) = 2\pi \left[ \frac{I_0(\tau^*)}{2} + \frac{I_1(\tau^*)}{3} \right], \tag{20}$$

$$F^+(\tau^*) = 2\pi \left[ \frac{I_0(\tau^*)}{2} - \frac{I_1(\tau^*)}{3} \right]. \tag{21}$$

This reads:

$$2\pi I(\tau^*, \pm\mu) = \frac{1}{2} \left[ (2 \pm 3\mu)F^-(\tau^*) + (2 \mp 3\mu)F^+(\tau^*) \right], \tag{22}$$

and therefore:

$$2\pi \int_0^1 I(\tau^*, \pm\mu)\mathrm{d}\mu = \frac{1}{4} \left[ (4 \pm 3)F^-(\tau^*) + (4 \mp 3)F^+(\tau^*) \right]. \tag{23}$$

Eventually:

$$\pi\omega^* \int_0^1 \int_{-1}^1 (1 \pm 3g^*\mu\mu')I(\tau^*, \mu')\mathrm{d}\mu'\mathrm{d}\mu = \frac{\omega^*}{4} \left[ (4 \pm 3g^*)F^-(\tau^*) + (4 \mp 3g^*)F^+(\tau^*) \right]. \tag{24}$$





Substituting Eqs 23 and 24 into Eqs 16 and 17 we obtain:

$$\frac{dF^-(\tau^*)}{d\tau^*} = -\frac{1}{4}\left[7F^-(\tau^*) + F^+(\tau^*)\right] + \frac{\omega^*}{4}\left[(4+3g^*)F^-(\tau^*) + (4-3g^*)F^+(\tau^*)\right] + \omega^*\gamma_4 F_0 e^{-\tau^*/\mu_0}, \tag{25}$$

$$\frac{dF^+(\tau^*)}{d\tau^*} = \frac{1}{4}\left[F^-(\tau^*) + 7F^+(\tau^*)\right] - \frac{\omega^*}{4}\left[(4-3g^*)F^-(\tau^*) + (4+3g^*)F^+(\tau^*)\right] - \omega^*\gamma_3 F_0 t e^{-\tau^*/\mu_0}., \tag{26}$$

which can be factorized as:

$$\frac{dF^-(\tau^*)}{d\tau^*} = \gamma_2 F^+(\tau^*) - \gamma_1 F^-(\tau^*) + \omega^*\gamma_4 F_0 e^{-\tau'/\mu_0}, \tag{27}$$

$$\frac{dF^+(\tau^*)}{d\tau^*} = \gamma_1 F^+(\tau^*) - \gamma_2 F^-(\tau^*) - \omega^*\gamma_3 F_0 e^{-\tau'/\mu_0}. \tag{28}$$

where

$$\gamma_1 = \frac{1}{4}\left[7 - \omega^*(4+3g^*)\right] \quad \text{and} \quad \gamma_2 = -\frac{1}{4}\left[1 - \omega^*(4-3g^*)\right]. \tag{29}$$

This corresponds to two coupled first-order differential equations, with matrix $\mathcal{A}$ such that:

$$\mathcal{A} = \begin{pmatrix} -\gamma_1 & \gamma_2 \\ -\gamma_2 & \gamma_1 \end{pmatrix}, \tag{30}$$

which has two eigenvalues $k_e$ and $-k_e$ with $k_e = \sqrt{\gamma_1^2 - \gamma_2^2}$, and corresponding eigenvectors:

$$v_1 = \begin{pmatrix} 1 \\ 1/\Gamma \end{pmatrix} \quad \text{and} \quad v_2 = \begin{pmatrix} 1 \\ \Gamma \end{pmatrix}, \tag{31}$$

where $\Gamma = \frac{\gamma_1 - k_e}{\gamma_2}$. A particular solution of this system is sought as:

$$F_p^-(\tau^*) = G^- e^{-\tau^*/\mu_0}, \tag{32}$$

$$F_p^+(\tau^*) = G^+ e^{-\tau^*/\mu_0}. \tag{33}$$

Inserting these expressions into Eqs 27 and 28 results in two equations with two unknowns which gives $G^-$ and $G^+$:

$$G^- = \frac{\mu_0^2 \omega^* F_0}{(k_e\mu_0)^2 - 1}\left[(\gamma_1 + 1/\mu_0)\gamma_4 + \gamma_2\gamma_3\right], \tag{34}$$

$$G^+ = \frac{\mu_0^2 \omega^* F_0}{(k_e\mu_0)^2 - 1}\left[(\gamma_1 - 1/\mu_0)\gamma_3 + \gamma_2\gamma_4\right]. \tag{35}$$

This overall gives the following solutions for the total (i.e. sum of direct and diffuse) downward and upward fluxes :

$$F_{\text{tot}}^-(\tau^*) = A e^{-k_e\tau^*} + B e^{k_e\tau^*} + (G^- + \mu_0 F_0)e^{-\tau^*/\mu_0}, \tag{36}$$

$$F_{\text{tot}}^+(\tau^*) = \Gamma A e^{-k_e\tau^*} + \frac{B}{\Gamma} e^{k_e\tau^*} + G^+ e^{-\tau^*/\mu_0}. \tag{37}$$





Note that these above solutions are consistent with those used by Toon et al. (1989). In addition to the fluxes, the actinic flux can be derived:

$$F_{\text{act}}(\tau^*) = 2\pi \int_{-1}^{1} I(\tau^*, \mu)\mathrm{d}\mu. \tag{38}$$

Given Eq. 19, and considering the contribution of the direct radiation to the actinic flux, this overall reads:

$$F_{\text{act}}(\tau^*) = 2(F^-(\tau^*) + F^+(\tau^*)) + F_0 e^{-\tau^*/\mu_0}. \tag{39}$$

## 2.4 Alternative formulation to match the AART theory

The form of Eqs 36 and 37 is common to all two-stream methods (e.g. Meador and Weaver, 1980). Considering a semi-infinite snowpack, it is clear that $\Gamma$ corresponds to the diffuse albedo and $k_e$ to the asymptotic flux extinction coefficient. To allow a perfect match of the two-stream solution with the AART theory in the case of a single layer we propose to test a new variant denoted TARTES∞AART in the following, that uses alternative expressions for $\Gamma$, $k_e$, $G^-$ and $G^+$. More specifically, following Kokhanovsky and Zege (2004) we let:

$$\Gamma = \exp\left(-4\sqrt{\frac{1-\omega}{3(1-g)}}\right), \tag{40}$$

$$k_e = \sqrt{3(1-\omega)(1-g)}. \tag{41}$$

The parameters $G^-$ and $G^+$ are chosen so that the direct albedo of the AART theory is obtained in the case of a semi-infinite snowpack, which is given by:

$$\alpha_{\text{dir}}(\mu_0) = \exp\left(-\frac{12}{7}(1+2\mu_0)\sqrt{\frac{1-\omega}{3(1-g)}}\right). \tag{42}$$

It implies that:

$$\Gamma A + G^+ = \alpha_{\text{dir}}\mu_0 F_0. \tag{43}$$

We also have $A + G^- = 0$ because the incident diffuse radiation is zero, but we need another constraint on $G^-$ and $G^+$. We impose that their sum be equal to that of the $\delta$-Eddington approximation, that is:

$$G^- + G^+ = \frac{3}{2}G_0\left(1 + g(1-\omega)\right), \tag{44}$$

where:

$$G_0 = \frac{\mu_0^2\omega F_0}{(k_e\mu_0)^2 - 1}. \tag{45}$$

This finally reads:

$$G^- = \frac{\frac{3}{2}G_0\left(1 + g(1-\omega)\right) - \alpha_{\text{dir}}\mu_0 F_0}{\Gamma + 1}, \tag{46}$$

$$G^+ = \frac{3}{2}G_0\left(1 + g(1-\omega)\right) - G^-. \tag{47}$$





When these new formulas are used $\tau^*$ in the previous Eqs 11–39 must be changed to $\tau$, without $\delta$-scaling.

Note that we also tested the $\delta$-scaling with these AART formulas, and found similar performances as the present TARTES∞AART formulas without scaling. These results are not reported.

## 2.5 Extension to a multi-layered snowpack

The equations derived so far all considered a homogeneous layer. For a multi-layered snowpack, the fluxes within each layer of the snowpack have the general form given by Eqs 36 and 37 but to determine the actual fluxes, the constants $A$ and $B$ should be determined for each layer, which amounts to $2N$ unknowns ($A_i, B_i, i \in \{1, n\}$) for a snowpack with $N$ layers. These unknowns are deduced from the continuity of $F_{\text{tot}}^{\pm}(\tau^*)$ at the layer interfaces ($2(N-1)$ equations) and the top and bottom boundary conditions (2 equations). Continuity of the diffuse fluxes at $\tau_i^*$ between layers $i$ and $i+1$ reads:

$$A_i e^{-k_{e,i}\tau_i^*} + B_i e^{k_{e,i}\tau_i^*} + G_i^- e^{-\tau_i^*/\mu_0} = A_{i+1} e^{-k_{e,i+1}\tau_i^*} + B_{i+1} e^{k_{e,i+1}\tau_i^*} + G_{i+1}^- e^{-\tau_i^*/\mu_0}, \tag{48}$$

$$\Gamma_i A_i e^{-k_{e,i}\tau_i^*} + \frac{B_i}{\Gamma_i} e^{k_{e,i}\tau_i^*} + G_i^+ e^{-\tau_i^*/\mu_0} = \Gamma_{i+1} A_{i+1} e^{-k_{e,i+1}\tau_i^*} + \frac{B_{i+1}}{\Gamma_{i+1}} e^{k_{e,i+1}\tau_i^*} + G_{i+1}^+ e^{-\tau_i^*/\mu_0}. \tag{49}$$

From now on we use the notation $k_i = k_{e,i}$ and define $A_i' = A_i e^{-k_i \tau_{i-1}^*}$ and $B_i' = B_i e^{k_i \tau_{i-1}^*}$. Note also that $\tau_0 = 0$. The boundary conditions at the top of the snowpack, where the diffuse flux is $F_0^{\text{diff}}$, and at the bottom, where the underlying surface is assumed lambertian and characterized by its albedo $\alpha_b$, read:

$$A_1 + B_1 + G_1^- = F_0^{\text{diff}} \tag{50}$$

$$\Gamma_N A_N e^{-k_N \tau_N^*} + \frac{B_N}{\Gamma_N} e^{k_N \tau_N^*} + G_N^+ e^{-\tau_N^*/\mu_0} = \alpha_b \left( A_N e^{-k_N \tau_N^*} + B_N e^{k_N \tau_N^*} + (G_N^- + \mu_0 F_0) e^{-\tau_N^*/\mu_0} \right). \tag{51}$$

The linear system formed by these $2N$ independent equations can be written as:

$$MX = V. \tag{52}$$

To avoid extremely large and low values in the matrix $M$ we incorporate the exponential terms in the vector $X$, so that we
have:

$$X = {}^t(A_{i,i\in\{1,n\}}'). \tag{53}$$



Accordingly the matrix $M$ reads:

$$
\begin{pmatrix}
1 & 1 & 0 & 0 & 0 & .. & 0 & 0 \\
e_1^- & e_1^+ & -1 & -1 & 0 & .. & 0 & 0 \\
\Gamma_1 e_1^- & \frac{1}{\Gamma_1}e_1^+ & -\Gamma_2 & -1/\Gamma_2 & 0 & .. & 0 & 0 \\
0 & 0 & e_2^- & e_2^+ & .. & .. & 0 & 0 \\
0 & 0 & \Gamma_2 e_2^- & \frac{1}{\Gamma_2}e_2^+ & .. & .. & 0 & 0 \\
0 & 0 & 0 & 0 & .. & .. & 0 & 0 \\
.. & .. & .. & .. & .. & .. & .. & .. \\
0 & 0 & 0 & 0 & 0 & .. & -1 & -1 \\
0 & 0 & 0 & 0 & 0 & .. & -\Gamma_N & -1/\Gamma_N \\
0 & 0 & 0 & 0 & 0 & .. & (\Gamma_N - \alpha_b)e_N^- & (1/\Gamma_N - \alpha_b)e_N^+
\end{pmatrix},
\tag{54}
$$

and:

$$
V = {}^t(F_0^{\text{diff}} - G_1^-, .., dG_i^- e^{-\tau_i^*/\mu_0}, dG_i^+ e^{-\tau_i^*/\mu_0}, ..., \left[\alpha_b(G_N^- + \mu_0 F_0) - G_N^+\right] e^{-\tau_N^*/\mu_0}),
\tag{55}
$$

where we used the notation $e_i^\pm = e^{\pm k_i d\tau_i^*}$ and $d\tau_i^* = \tau_i^* - \tau_{i-1}^*$ is the optical depth of layer $i$ and $dG_i^\pm = G_{i+1}^\pm - G_i^\pm$. The matrix $M$ can be tridiagonalized doing consecutively the following replacement operations on the lines $M_j$ of $M$ for $2 \leq j < N$ even:

1. $M_j - \Gamma_{i/2+1}L_{j+1} \to M_j$,

2. $(1 - \Gamma_{i/2}\Gamma_{i/2+1})M_{j+1} - \Gamma_{i/2}M_j \to M_{j+1}$.

The new matrix $M$ reads:

$$
\begin{pmatrix}
1 & 1 & 0 & 0 & 0 & .. & 0 & 0 \\
(1-\Gamma_1\Gamma_2)e_1^- & (1-\Gamma_2/\Gamma_1)e_1^+ & (\Gamma_2^2-1) & 0 & 0 & .. & 0 & 0 \\
0 & (1/\Gamma_1-\Gamma_1)e_1^+ & (\Gamma_1-\Gamma_2) & (\alpha_1-1/\alpha_2) & 0 & .. & 0 & 0 \\
0 & 0 & (1-\Gamma_2\Gamma_3)e_2^- & (1-\Gamma_3/\Gamma_2)e_2^+ & .. & .. & 0 & 0 \\
0 & 0 & 0 & (1/\Gamma_2-\Gamma_2)e_2^+ & .. & .. & 0 & 0 \\
0 & 0 & 0 & 0 & .. & .. & 0 & 0 \\
.. & .. & .. & .. & .. & .. & .. & .. \\
0 & 0 & 0 & 0 & 0 & .. & (\Gamma_N^2-1) & 0 \\
0 & 0 & 0 & 0 & 0 & .. & (\Gamma_{N-1}-\Gamma_N) & (\Gamma_{N-1}-1/\Gamma_N) \\
0 & 0 & 0 & 0 & 0 & .. & (\Gamma_N-\alpha_b)e_N^- & (1/\Gamma_N-\alpha_b)e_N^+
\end{pmatrix}.
$$

Accordingly the new vector $V$ reads:

$$
V = {}^t(F_0^{\text{diff}} - G_1^-, .., (dG_i^- - \Gamma_{i+1}dG_i^+)e^{-\tau_i^*/\mu_0}, (dG_i^+ - \Gamma_i dG_i^-)e^{-\tau_i^*/\mu_0}, ..., \left[\alpha_b(G_N^- + \mu_0 F_0) - G_N^+\right] e^{-\tau_N^*/\mu_0}).
\tag{56}
$$





The $2N$ unknowns are efficiently retrieved by inversion of the tridiagonal system. Then the fluxes at each interface are calcu-
lated as follows:

$$F_{\text{tot}}^-(\tau_i^*) = A_i' e^{-k_i^* d\tau_i^*} + B_i' e^{k_i^* d\tau_i^*} + (G_i^- + \mu_0 F_0) e^{-\tau_i^*/\mu_0}, \tag{57}$$

$$F_{\text{tot}}^+(\tau_i^*) = \Gamma_i A_i' e^{k_i^* d\tau_i^*} + \frac{B_i'}{\Gamma} e^{k_i^* d\tau_i^*} + G_i^+ e^{-\tau_i^*/\mu_0}. \tag{58}$$

Note that in practice the fluxes can be computed at any requested depth. To this end, the layer $i$ corresponding to this depth is
first identified. The ratio of the distance between the above interface and the requested depth, and the thickness of the layer, is
used to scale $d\tau_i^*$ in the above solutions.

## 2.6 Computed quantities

Beyond the fluxes (and actinic fluxes) that are the native variables returned by the above equations it is possible to compute the
energy absorbed by layer $i$ as:

$$E_i = \underbrace{F_{\text{tot}}^+(\tau_i^*) - F_{\text{tot}}^+(\tau_{i-1}^*)}_{E_u} - \underbrace{\left(F_{\text{tot}}^-(\tau_i^*) - F_{\text{tot}}^-(\tau_{i-1}^*)\right)}_{E_d} \tag{59}$$

$$E_u = \Gamma_i A_i'(e^{-k_i^* d\tau_i^*} - 1) + \frac{B_i'}{\Gamma_i}(e^{k_i^* d\tau_i^*} - 1) + G_i^+(e^{-\tau_i^*/\mu_0} - e^{-\tau_{i-1}^*/\mu_0}) \tag{60}$$

$$E_d = A_i'(e^{-k_i^* d\tau_i^*} - 1) + B_i'(e^{k_i^* d\tau_i^*} - 1) + (G_i^- + \mu_0 F_0)(e^{-\tau_i^*/\mu_0} - e^{-\tau_{i-1}^*/\mu_0}). \tag{61}$$

while the energy absorbed by the ground is given by:

$$E_{\text{bottom}} = (1 - \alpha_b)(A_N' e_N^- + B_N' e_N^+ + (G_N^- + \mu_0 F_0) e^{-\tau_N^*/\mu_0}). \tag{62}$$

The albedo of the snowpack is also calculated as the ratio of the upward to the downward flux at the surface:

$$\alpha = \frac{1}{\mu_0 F_0 + F_0^{\text{diff}}} \left( \Gamma_1 A_1 + \frac{B_1}{\Gamma_1} + G_1^+ \right). \tag{63}$$

So far we have not specified anything about the spectral dimension of incident light. Implicitly all above derivations are valid
for monochromatic radiation, so that TARTES is by essence a monochromatic model. Since the single scattering properties of
the snowpack are wavelength-dependent, the matrix $M$ and the vector $V$ are computed at each relevant wavelength. Broadband
quantities are thus obtained by summing the contribution of all wavelengths. For instance the broadband albedo $\overline{\alpha}$ is obtained
through spectral integration:

$$\overline{\alpha} = \frac{\sum_1^N \alpha(\lambda_i)(F_0^{\text{diff}}(\lambda_i) + \mu_0 F_0(\lambda_i))}{\sum_1^N F_0^{\text{diff}}(\lambda_i) + \mu_0 F_0(\lambda_i)}. \tag{64}$$



## 2.7 Treatment of diffuse incident radiation

As seen previously, the incident radiation in TARTES can be direct or diffuse. Wiscombe (1977) has shown that in the case of diffuse radiation the performances of the $\delta$-Eddington approximation were limited, sometimes leading to negative values of albedo. This is why Warren and Wiscombe (1980) computed the diffuse albedo as an angular average of direct albedos. In TARTES also, the most accurate strategy to handle diffuse radiation is to compute the integrated sum of direct radiation coming from all directions, following an angular distribution such that $p(\theta_0) = \cos\theta_0$. Hence it requires integrating the solutions for direct incident light at various angles. As only the vector $V$ depends on incident light characteristics, to compute the optical properties of a snowpack at various angles of incidence, $M$ has to be calculated only once, which is computationally relatively efficient.

An alternative strategy proposed in TARTES is to consider that diffuse radiation can be approximated by direct radiation at an effective zenith angle $\theta_{\mathrm{diff}}$ such that (see Eq. 42):

$$\frac{3}{7}(1 + 2\cos\theta_{\mathrm{diff}}) = 1, \tag{65}$$

which corresponds approximately to an angle of $48.2°$. This alternative is the default option in TARTES (hereinafter referred to as $48.2°$). Note that in the initial version of TARTES (Libois et al., 2013) the diffuse albedo was calculated using a direct component at an angle of $53°$ based on equivalence tests using DISORT-Mie. This angle was changed to the approximate value of $48°$ on 22 June 2022, and is now obtained with the exact calculation (Eq. 65).

Note that despite the problem identified by Wiscombe (1977), the pure diffuse boundary condition of the two-stream method is implemented in TARTES (hereinafter denoted "2S") and should be selected for testing only as done in Sec. 4.1.1. To avoid negative albedo, we set $\alpha = 0$ when a negative value is obtained, which occurs when the ice absorption is very large (1400–1600 nm) (Sec. 4.1.1).

## 2.8 Treatment of optically deep layers and snowpacks

When a layer is too thick, the terms $e_i^{\pm}$ become either extremely large or small, and in both cases cannot be handled numerically. To avoid this, when a layer is too thick (practically when $k_i d\tau_i^* > 200$), its optical depth is modified so that $k_i d\tau_i^* = 200$. In addition, When a snowpack is very deep, energy does not penetrate through the whole snowpack, it is essentially absorbed in the topmost layers. To save computation time, the snowpack used for the calculations is reduced to the top $n$ layers, where $n$ is the smallest integer such that:

$$\sum_1^n k_i^* d\tau_i > 30. \tag{66}$$

At the same time, the optical thickness of the last layer is set to $30/k_n$ and the underlying albedo is set to $1$ to ensure the underlying surface does not absorb energy.





## 2.9 Single scattering properties of snow

The previous sections detailed how the fluxes and vertical profiles of absorbed energy within a multi-layered snowpack are computed. This section details how the single scattering properties of snow, namely $\sigma_e$, $\omega$ and $g$, are determined from the snow physical properties, namely SSA, density, grain shape and impurity contents. It is worth having in mind that the RTE applies to a continuous medium. As snow is a porous medium, it is common to define an optically equivalent continuous medium to represent it. In practice its extinction coefficient is determined from the number concentration $N$ (m$^{-3}$) of snow grains and the average extinction cross section of snow grains (Kokhanovsky and Zege, 2004):

$$\sigma_e = N\overline{C_{\text{ext}}} \tag{67}$$

This strategy, that has been originally developed to compute for instance the optical properties of clouds (Stephens, 1978), implicitly assumes that scatterers are independent. Although this is unlikely to be the case in snow which is a dense medium, this formalism remains widely used since it has proved its efficiency to simulate snow optical properties in the solar spectrum where ice absorption is relatively low. Actually it remains efficient as long as the asymmetry factor is also computed assuming independent scatterers (Kokhanovski, 2004). Using this representation, snow density is related to the average volume of snow grains $\overline{V}$: $\rho = N\rho_{\text{ice}}\overline{V}$. We further assume that snow grains are large compared to the wavelength of solar radiation, so that $C_{\text{ext}} = 2\Sigma$ (where $\Sigma$ is the projected area of an individual grain) and that the grains are convex so that $\Sigma = S/4$, where $S$ is the total surface area of a grain. Hence $\sigma_e$ finally reads:

$$\sigma_e = \frac{\rho\text{SSA}}{2}, \tag{68}$$

where SSA equals by definition $\dfrac{\overline{S}}{\rho_{\text{ice}}\overline{V}}$. Note that this expression was originally known for convex particles only (e.g. Libois et al., 2013) but was recently applied to a more-general porous medium (Malinka, 2014).

The single scattering albedo is computed after Kokhanovsky and Macke (1997), who propose an analytical expression depending on the refractive index $m = n - i\chi$ and grain shape $\mathcal{S}$, based on Monte Carlo computations relying on the geometrical optics approximation:

$$(1 - \omega) = \frac{1}{2}(1 - W(n))(1 - e^{-\psi(n,\mathcal{S})c}), \tag{69}$$

where

$$c = \frac{24\pi\chi}{\rho_{\text{ice}}\lambda\text{SSA}}, \tag{70}$$

and $\rho_{\text{ice}}$ is the bulk density of ice. $W$ does not depend on grain shape (for randomly oriented, convex, particles, Kokhanovsky and Macke, 1997) and is assumed to depend linearly on $n$ based on tabulated values in (Kokhanovski, 2004, p. 61), so that in TARTES:

$$W(n) = 0.0611 + 0.17(n - 1.3). \tag{71}$$





Likewise,

$$\psi(n,\mathcal{S}) = \frac{2}{3}\frac{B(n,\mathcal{S})}{1-W(n)}. \tag{72}$$

where $B(n,\mathcal{S})$ is the absorption enhancement parameter. Note that at low absorption Eq. 69 collapses to equation 6 of Libois et al. (2013). TARTES proposes three options to compute $B$. The first option (the only one in previous version) is a linear dependence on $n$ based on Kokhanovsky and Macke (1997), so that:

$$B(n,\mathcal{S}) = B_0(\mathcal{S}) + 0.4(n-1.3), \tag{73}$$

where $B_0(\mathcal{S})$ can be prescribed by the user to account for a particular geometrical shape. For instance, for spherical particles $B_0 = 1.25$ (Libois et al., 2013). The second option is

$$B(n,\mathcal{S}) = n^2 \tag{74}$$

which stems from the recent work on random media and is now the recommended and default option (Malinka, 2014; Robledano et al., 2023). The last option gives the possibility to the user to prescribe a constant value or a value per wavelength.

The asymmetry factor $g$ also depends on the detailed snow microstructure, but in the granular representation it can be computed from the scattering phase function of individual grains. At weakly absorbing wavelengths $g$ mainly depends on snow grain shape $\mathcal{S}$ but at absorbing wavelengths it also depends on the ice imaginary part of the refractive index $\chi$ and SSA. In TARTES $g$ is calculated in consistency with $B$, according to the three options. In the first option $g(n,\mathcal{S})$ is computed after Kokhanovsky and Macke (1997):

$$g(n,\mathcal{S}) = g_\infty(n) - [g_\infty(n) - g_0(n,\mathcal{S})]e^{-y(n,\mathcal{S})c}. \tag{75}$$

$g_\infty(n)$ is the asymmetry factor of a purely absorbing sphere and $g_0(n,\mathcal{S})$ is the asymmetry factor of the non absorbing particle of shape $\mathcal{S}$. $g_0$ and $g_\infty$ are both assumed to depend linearly on $n$, so that:

$$g_\infty(n) = 0.9751 - 0.105(n-1.3), \tag{76}$$

$$g_0(n,\mathcal{S}) = g_0(\mathcal{S}) - 0.38(n-1.3), \tag{77}$$

where $g_0(\mathcal{S})$ is prescribed by the user. Again the dependence on $n$ corresponds to that of spheres (Kokhanovski, 2004). In the second option, $g = 0.82$ (Robledano et al., 2023) and in the third option, the user prescribes the value as a constant or as one value per wavelength.

Finally $y$ is also assumed to depend linearly on $n$. In TARTES the expression corresponding to spheres is taken so that (Kokhanovski, 2004):

$$y(n) = 0.728 + 0.752(n-1.3). \tag{78}$$

Note that for the three options, the variables $W$, $g_\infty$ and $y$ are calculated by linear relationships (Eqs 71, 76 and 78) corresponding to spheres. This may result in inconsistencies with respect to $B$ and $g$, but given that at present these three former variables have not been investigated for snow we prefer to keep the relationship used up to now.




## 2.10 Impurities

At the wavelengths ice is very weakly absorbing, the optical properties of snow are very sensitive to the presence of light absorbing impurities. TARTES can account for such impurities, in practice black carbon (BC), dust and humic-like substances (HULIS). For sake of simplicity it is assumed that impurities are external to snow grains. When impurities are added in realistic, low quantities $N_i$, we assume that the extinction coefficient of snow is unchanged but the absorption coefficient is altered. It follows that the single scattering co-albedo is:

$$(1 - \omega) = (1 - \omega)_{\text{snow}} + \frac{1}{\sigma_e} \sum_i N_i C_{\text{abs}}^i, \tag{79}$$

where $C_{\text{abs}}^i$ is the average absorption cross section of impurities $i$. Rewritten as a function of the bulk mass concentration $c_i$ (kg kg$^{-1}$) and using Eq. 69, it reads:

$$(1 - \omega) = \frac{1}{2}(1 - W(n))(1 - e^{-\psi(n,s)c}) - \frac{2}{\lambda \text{SSA}} \sum_i \text{MAE}^i c_i, \tag{80}$$

where $\text{MAE}^i$ is the mass absorption efficiency (in m$^2$ kg$^{-1}$, e.g. Caponi et al., 2017).

To calculate MAE, TARTES uses different formulations according to the particle size. For small particles compared to the wavelength (applies to BC and HULIS), the absorption cross section $C_{\text{abs}}$ of an impurity of type $i$, of volume $V_i$ and complex refractive index $m_i$ is given by Kokhanovski (2004):

$$C_{\text{abs}}^i = -\frac{6\pi V_i}{\lambda} \text{Im}\left(\frac{m_i^2 - 1}{m_i^2 + 2}\right). \tag{81}$$

Dividing by $\rho_i V_i$, where $\rho_i$ is the impurity bulk density, yields the mass absorption efficiency:

$$\text{MAE}^i = -\frac{6\pi}{\lambda \rho_i} \text{Im}\left(\frac{m_i^2 - 1}{m_i^2 + 2}\right). \tag{82}$$

In TARTES, the characteristics of BC can be taken from Bond and Bergstrom (2006) ($\rho_{\text{BC}} = 1800$ kg m$^{-3}$ and $m_{\text{BC}} = 1.95 - 0.79i$) or by default from SNICAR-ADv3 (Flanner et al., 2021). The values for the HULIS are taken from Hoffer et al. (2006).

For large particles (applies to dust), the absorption is not simply related to the imaginary part of their refractive index and the volume, but depends on the shape, size and other impurities particularities. While Mie theory applies to spherical particles – and other more complex theories to more general shapes (Mishchenko et al., 1996) – the computation is usually intensive. Instead TARTES directly uses tabulated MAE values that can be obtained from independent calculations or from *in situ* measurements. More precisely, the MAE is calculated from the MAE at a specific wavelength (usually $\lambda_0 = 400$ nm or $\lambda_0 = 550$ nm) and the spectral dependence given by the Angström absorption exponent AAE such as:

$$\text{MAE}(\lambda) = \text{MAE}(\lambda_0)\left(\frac{\lambda}{\lambda_0}\right)^{-\text{AAE}}. \tag{83}$$

In TARTES v2.0, values from Caponi et al. (2017) obtained from different regions in the world are implemented. For small particles (PM2.5), available locations are Libya, Morocco, Algeria, Mali, Saudi Arabia, Kuwait, Namibia, China, Australia,





and for large particles (PM10): Libya, Algeria, Bodele, Saudi Arabia, Namibia, China, Arizona, Patagonia, Australia(Caponi
et al., 2017).

# 3   Numerical implementation

TARTES was initially implemented in Python, and then converted in part to Fortran to be integrated in the detailed snowpack
model Crocus (Vionnet et al., 2012), where it can be used to compute the profile of absorption in the snowpack (Libois et al.,
2015; Tuzet et al., 2017). Both versions are based on the same equations, though the Python version has more features beyond
absorption calculation. Note also that the Fortran version is not systematically updated along with the Python version.

## 3.1   Python version

TARTES v2.0 is compatible with Python 3.7 and higher. The four main entry points for the user are the functions *albedo*,
*absorption_profile*, *irradiance_profile*, *actinic_profile*. They take as inputs the wavelengths at which the computation is per-
formed, the properties of each layer (thickness, SSA, density, $B_0$ and $g_0$, impurities), the name of the ice refractive index
database (Warren and Brandt, 2008; Picard et al., 2016), the bottom albedo, and the illumination conditions (solar zenith an-
gle and fraction of direct radiation). The outputs are the albedo, the absorption, the up and downwelling irradiances, and the
actinic flux for each wavelength, for each function respectively. In addition, a function allows the calculation of broadband
albedo given the incident spectrum distribution.

These user functions internally call the core function *tartes* that computes the intrinsic optical properties of each layer from
their physical properties and impurities contents, and then call *two_stream_rt* which solves the radiative transfer equation,
eventually allowing to compute all the quantities mentioned here above (Fig. 1).

The code has a test suite (10 tests currently), that can be automatically run using the software *pytest* to check the conformity
of the code.

## 3.2   Fortran version

The original TARTES code was converted in a Fortran version suitable for the integration in Crocus (Vionnet et al., 2012).
This model predicts the evolution of a multi-layered snowpack from local meteorological forcings. For this the absorption
of the solar energy in each layer needs to be computed at each time step. This is critical to compute the energy budget of
the snowpack and consequently the temperature profile and gradients that control snow metamorphism (Flanner and Zender,
2005). The original optical scheme in Crocus (Brun et al., 1989) estimates the albedo, hence the total absorbed energy, in
three spectral bands from empirical relationships with the grain shape and size, and then applies an *ad hoc* Beer-Lambert
law to distribute the absorbed energy in the layers. Using a proper radiative transfer model has many advantages: the higher
spectral resolution (10 nm by default) allows to resolve the spectral features of snow without resorting to questionable spectral
averages, it computes the profile of energy in a way that is consistent with the albedo, the direct and diffuse fluxes and the solar
zenith angle dependence are treated properly, absorption by the soil, snow grain shape and impurities are fully accounted for.



However, the computation time can be significant, especially if the spectral integration is highly resolved. This is the reason why studies have developed optimized wavelength sampling strategies to reduce the number of computations needed (van Dalum et al., 2019; Veillon et al., 2021).

Although the translation of TARTES in Fortran was intially motivated by the integration in Crocus, it is a self-contained model that can be integrated into any other model. The source code is part of SURFEX (http://www.umr-cnrm.fr/surfex/, last access: 1 Mar 2024, Masson et al., 2013). We call this version TARTES.F hereafter.

In terms of performance, a simulation for a two-layer snowpack at 106 wavelengths repeated 10,000 times takes 3.4 s with TARTES.F on a commodity laptop, while the Python version takes 148 s, about 40 times longer.

## 3.3   Related software and libraries for snow optics

### 3.3.1   SnowTARTES web application

SnowTARTES is an interactive web application (https://snow.univ-grenoble-alpes.fr/snowtartes, last access: 24 March 2024) meant to easily compute spectral albedo and irradiance profiles using TARTES without writing code. SnowTARTES uses the Python version, and thus provides exactly the same results. The snowpack is described layer by layer in a text form. In order to conduct a sensitivity analysis, any of the input parameters can be prescribed as a range (start, end, step) instead of a single value. These ranges are combined launching multiple calculations (limited to 10 maximum for the sake of visibility in the plot). The calculated albedo spectra are immediately plotted (one curve for each calculation) and can be downloaded as a Comma Separated Value formatted file.

SnowTARTES offers a selection of grain shapes to chose from, each one corresponding to a couple $(B_0, g_0)$ based on Libois et al. (2013) and Robledano et al. (2023). Similarly, it offers a selection of 7 background albedo spectra (grass, ice, several soils and 0% and 100% albedo) extracted from Johns Hopkins University Spectral Library (ECOSTRESS and formally ASTER library, https://speclib.jpl.nasa.gov/, last access: 24 March 2024).

### 3.3.2   Snowoptics python package

The python package Snowoptics (https://github.com/ghislainp/snowoptics, last access: 24 March 2024) provides a series of functions to compute albedo, extinction and BRDF for a semi-infinite homogeneous snowpack using the AART theory (Kokhanovsky and Zege, 2004; Kokhanovsky, 2012). The arguments of these functions are similar (name and unit) to TARTES. The package also provides functions to compute the effect of the slope on measured albedo and to correct from the slope effect based on (Picard et al., 2019) which is not available in TARTES. Here, Snowoptics is used in Section 4.2 to compare TARTES with AART.

### 3.3.3   AtmosRT python package

Because of the impact of the illumination geometry on the spectral albedo, and of the spectral distribution of incident irradiance on the broadband albedo, it is necessary to know the spectral solar irradiance. There are many available radiative transfer





models for the atmosphere that can provide this information for TARTES calculations, such as SMARTS (Gueymard, 2001), MODTRAN (Berk et al., 2014) or LibRadtran (Emde et al., 2016) but the offer in Python is limited. PyRTM is an unmaintained software package (https://github.com/Queens-Applied-Sustainability/PyRTM, last access: 24 March 2024) providing a Python 2 interface to access two general atmospheric radiative transfer models written in Fortran, namely SBDART (Ricchiazzi et al.,
1998) and SMARTS. From this, we developed the AtmosRT package, including support for Python 3 and a few additional minor improvements for ease of use. A function 'atmospheric_incident_spectrum' is implemented in TARTES to perform simple calculations with SBDART through AtmosRT, and directly provides the total flux and the direct fraction at each wavelength as required by the TARTES functions.

## 4 Results

Several simulations with TARTES and other models are compared in this section. By default, unless specified, we consider a semi-infinite homogeneous snowpack, i.e. made of a single layer, thick enough so that the bottom boundary does not influence the outgoing radiation and the presented profiles. The layer has an SSA of $20\,\mathrm{m^2 kg^{-1}}$ and the density is $350\,\mathrm{kg m^{-3}}$ but this latter variable has no influence on thick-snow albedo in the conventional radiative transfer framework used in TARTES (Malinka, 2023). Other particular conditions of the simulations are indicated for each case.

### 4.1 Self-consistency checks

TARTES has a few options to select between different approximations or different modes of calculations. Here, we compare these approximations, and check their consistency.

#### 4.1.1 Diffuse illumination

The three methods to take into account diffuse illumination in TARTES are compared in Fig. 2 for the default TARTES version
and in Fig. 3 for the TARTES∞AART version, for the spectral and broadband albedo of the default semi-infinite snowpack under typical illumination for wintertime Alpine snow. The reference method uses integration (noted "Integr.") of $n$ simulations in direct illumination mode with a solar zenith angle varying from 0 to 90° (with $n$=128 regular steps in cosine of the angle). The results show a close agreement with the direct calculations at a zenith angle of 48.2°, the equivalent angle obtained by matching the diffuse and direct expressions of AART albedo (Sect. 2.7). The latter approximation yields relatively accurate albedo, with
a deviation lower than 0.003 at wavelengths < 1300 nm, and never exceeding 0.007 in the investigated wavelength range (up to 2000 nm). The broadband albedo difference is virtually zero (0.0006). Similar results are obtained for the TARTES∞AART variant, with a maximum error of 0.011, and a broadband albedo (0.001).

In contrast, the formulation using diffuse illumination for the upper boundary condition of the two-stream approximation (2S method) performs poorly in the original TARTES, with deviations reaching nearly 0.02 at wavelengths < 1300 nm and
barely acceptable (> 0.03) beyond 1400 nm, although the broadband albedo difference of 0.0045 probably remains small enough for most applications. The inherent difficulty to handle diffuse radiation with the $\delta$-Eddington approximation has been



discussed extensively by Wiscombe (1977) and Wiscombe and Warren (1980). Conversely, the 2S method works well with the TARTES∞AART as it gives nearly equivalent results to the direct simulation at 48.2° (the dotted curves overlap in Fig 3b). This agreement is expected as TARTES∞AART was designed to be equivalent to AART, and 48.2° is the equivalent angle for

the direct and diffuse illumination in this theory.

As a conclusion, we suggest not to use the 2S method for practical applications, since it provides almost no benefit compared to the 48.2° method (at best an agreement, at worse a large error) over the spectral range of interest, and the calculation is only slightly faster than for a direct beam calculation (23 ms vs 34 ms for the computation in Fig. 2 involving 320 wavelengths).

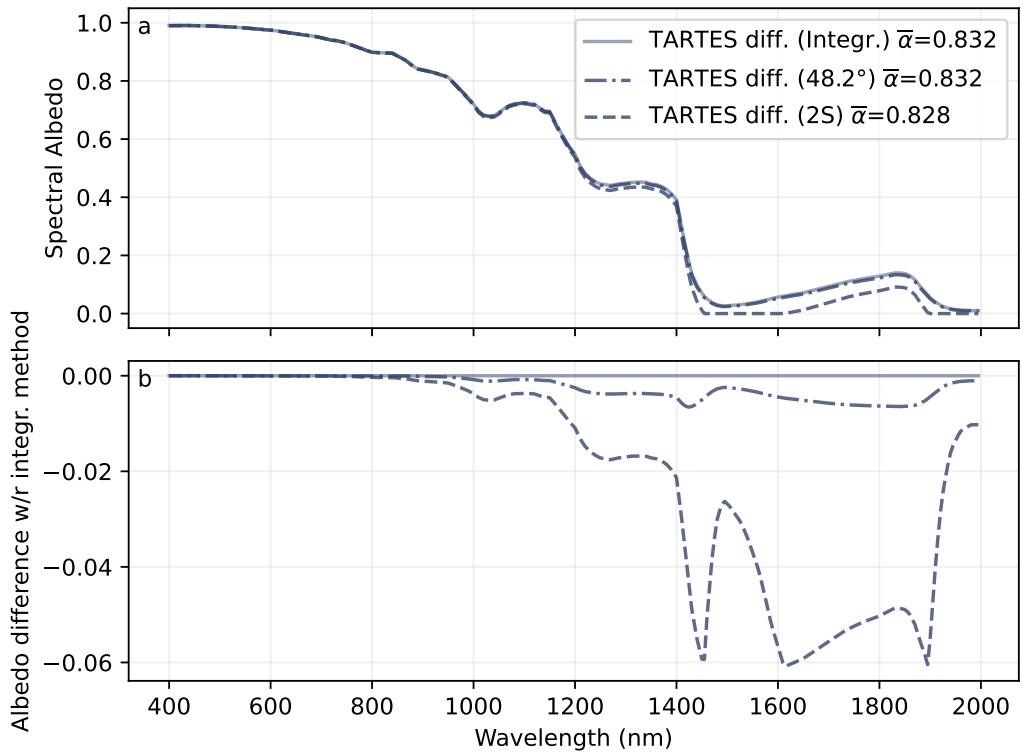

**Figure 2.** Comparison of albedo under diffuse illumination computed by three methods: 1) by integrating over all incident angles, 2) using a unique direct beam at the equivalent angle predicted by the AART (48.2°), and 3) using diffuse radiation for the upper boundary condition in the two-stream formulation (abbreviated 2S). The snowpack is semi-infinite with SSA = 20 $m^2 kg^{-1}$. The broadband albedo ($\omega$) is indicated in the legend for each spectrum.

For the profiles of irradiance, absorption and actinic flux only two methods are implemented, the integration and the direct

48.2° calculation. Figure 4 shows the profile of absorption close to the surface of the semi-infinite snowpack at 800 nm, a wavelength maximizing absorption since snow co-albedo (the proportion of absorbed radiation) is greater than at shorter wavelengths, and the incoming solar radiation is still relatively large compared to longer wavelengths. Only the original TARTES





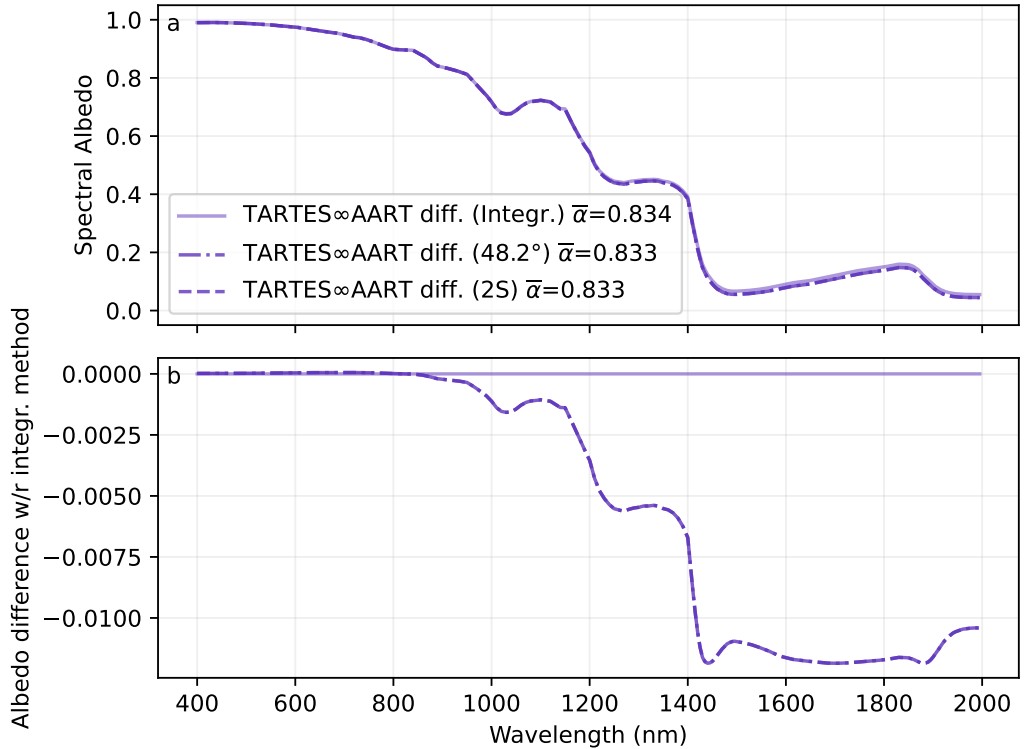

**Figure 3.** Same as Fig. 2 but for the TARTES∞AART variant.

version is used. At this wavelength, 90% of the absorption occurs in the topmost $\approx 4$ cm (Fig. 4a) for the snowpack considered here. The profiles of absorption obtained with both methods look similar, with maximum differences of about 1.5%, reached

close to the surface, as seen in Fig. 4b.

Based on these results, the direct 48.2° calculation was chosen as the default method to simulate diffuse radiation in TARTES. The integration is in principle more accurate but requires many more computations (solving the linear system for 128 angles instead of 1) even though measuring the execution time (51 ms instead of 34 ms) does not show a difference in the same proportion because only the constant vector of the linear system depends on the angle, not the matrix. In practice, users who

prefer the accuracy offered by the integration method can explicitly set this option.

### 4.1.2 Consistency between albedo, profile of irradiance and profile of absorption calculations

As the albedo, the irradiance and absorption profiles in snow and the absorption below the snowpack are computed by three distinct Python functions, it is worth checking that energy is conserved across these different quantities. To this end we first compare the albedo with the absorption profile. The semi-infinite snowpack is split in numerical layers of 1 cm (top first

meter), even though their properties are identical (SSA $= 20\,\mathrm{m^2\,kg^{-1}}$, density $= 350\,\mathrm{m^2\,kg^{-1}}$). Comparing the sum of all layer

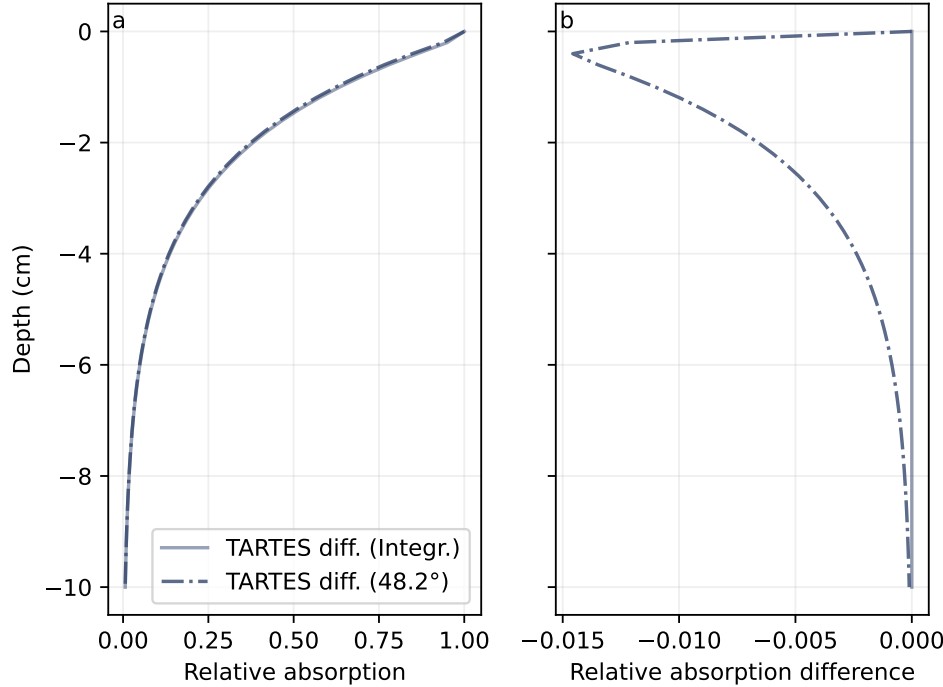

**Figure 4.** Comparison of the absorption profile under diffuse illumination computed by two methods: 1) by integrating over all incident angles and 2) using a unique direct beam at the equivalent angle predicted by the AART (48.2°). The snowpack is semi-infinite with SSA = $20 \, \mathrm{m^2 kg^{-1}}$.

absorption $A$ (divided by the incident irradiance, set arbitrarily to $1 \, \mathrm{W \, m^{-2}}$) with the co-albedo $1 - \alpha$, we found a residual numerical error $< 2 \times 10^{-16}$ for all wavelengths, which is close to the machine 64-bit floating point limit. Likewise, we checked that the calculated albedo perfectly matches the ratio of upwelling and downwelling irradiance at the surface, calculated from the vertical profiles of irradiance. These two tests are part of the automatic test suite available in the TARTES code base.

**4.2 Comparison of TARTES with the Asymptotic Analytical Radiative Transfer (AART)**

The comparison between TARTES and the AART to simulate the diffuse and direct albedos of a semi-infinite snowpack (a single thick homogeneous layer) is presented in Figs 5, 6 and 7.

Used with the same parameters (SSA, density, $B_0$ and $g_0$), the comparison between AART and TARTES∞AART for both the diffuse and direct illuminations demonstrates that the formulation in Eqs 40-41 and 46-47 allows our code to conform

to the AART analytical expression in the case of a semi-infinite homogeneous snowpack. The maximum error is indeed $4 \times$



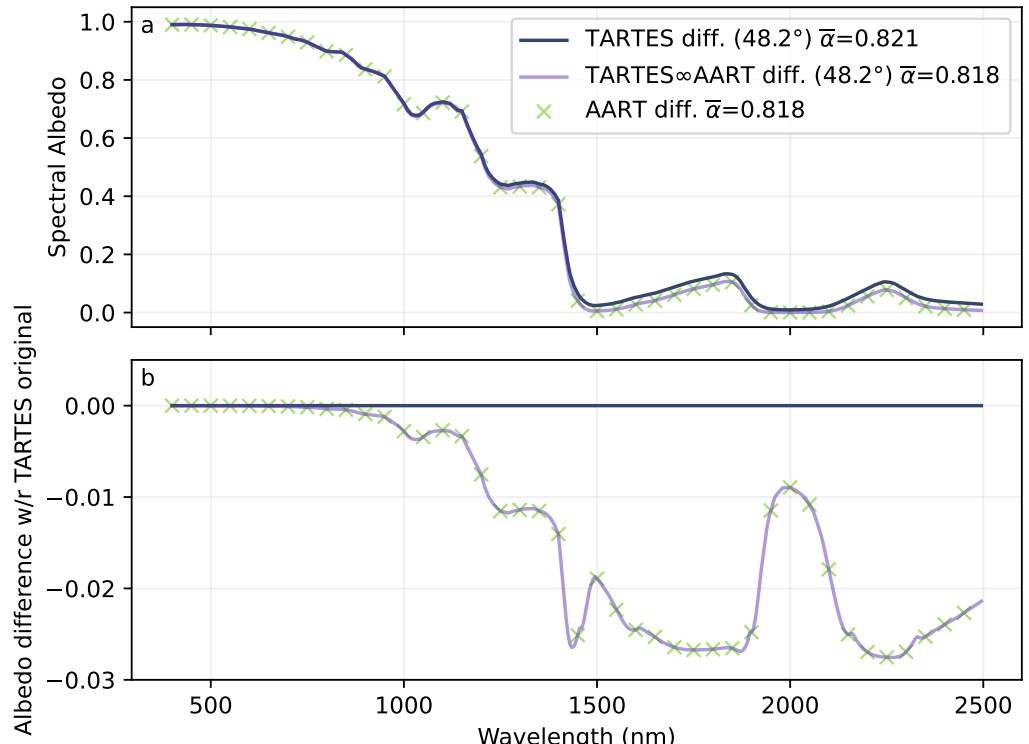

**Figure 5.** Comparison of the diffuse albedo spectra computed by TARTES (with 48.2°), TARTES∞AART (with 48.2°) and standalone AART under diffuse illumination. (a) shows the albedo and (b) the difference with respect to TARTES. The snowpack is semi-infinite with SSA = 20 m² kg⁻¹.

$10^{-16}$, corresponding to numerical rounding errors. This new formulation can be useful in cases the conformity with AART is essential.

AART is also indistinguishably similar to the original TARTES in the visible and up to 1400 nm on the spectrum plots in Figs 5a and 6a. However, the residuals in Figs 5b and 6b highlight that the differences are much larger than numerical rounding errors. Nonetheless, they remain <0.015 up to 1400 nm. At longer wavelengths the differences become noticeable and increase up to around 0.024. Interestingly, the differences between the models are mainly significant in the domain of strong ice absorption, where none of these models is expected to be valid. Indeed, AART is meant to be valid only for weak absorption (see fig. 8 of Kokhanovsky and Zege, 2004), and TARTES uses the $\delta$-Eddington approximation which likewise is only relevant for weak absorption (Wiscombe, 1977). Depending on the application, the reported errors in the longer wavelengths of the solar spectrum can be either negligible (e.g. broadband albedo calculations, absorption calculation) or major (e.g spectroscopic applications at 1550 nm as in Gallet et al., 2009).





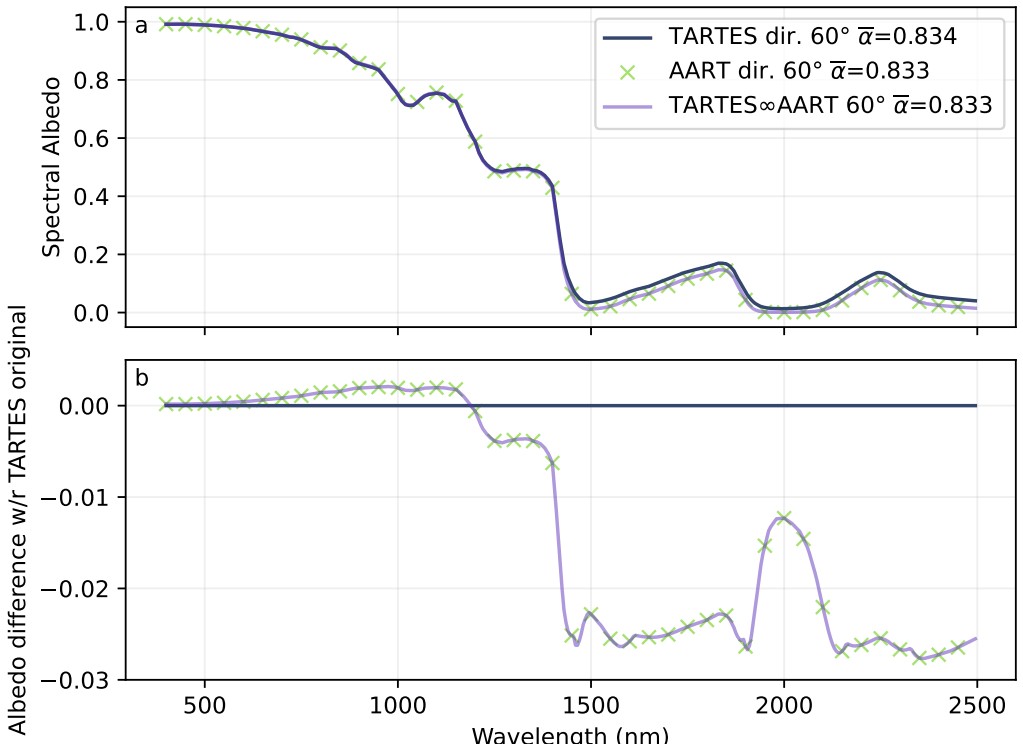

**Figure 6.** Same as Fig. 5, but for direct illumination (SZA=60°).

To further explore the difference between the models, Fig. 7 shows the albedo at 1300 nm as a function of SSA, $B_0$ and $g_0$. Again we observe the perfect similitude between AART and TARTES∞AART. On the other hand, the difference with original TARTES is small except for very low SSA where it reaches a maximum absolute value of 0.015. Such an error becomes

significant in relative error especially because albedo < 0.2. The differences when $B_0$ and $g_0$ are varied is even weaker over the investigated range. Exploring other wavelengths (results not shown) indicates that the difference is much smaller in the visible, and progressively increases in the near-infrared. These results obtained for the SSA and $B_0$, along with the wavelength dependence, again confirm that the models mainly diverge when the single scattering albedo is low (it is lower for large grains than for smaller ones), that is when the absorption is strong.

To provide a more general recommendation to future users, we explored a wide range of usual conditions ($\lambda$ <1400 nm, SSA in $5 - 100$ m$^2$ kg$^{-1}$, and SZA in 0-70°) and found a maximum difference of 0.025 (at SSA=5 m$^2$ kg$^{-1}$, SZA=40° and 1400 nm). Furthermore, 90% of the simulations show a very small difference <0.002, leading to the conclusion that, for a semi-infinite snowpack, TARTES is virtually equivalent to AART in most usual conditions.



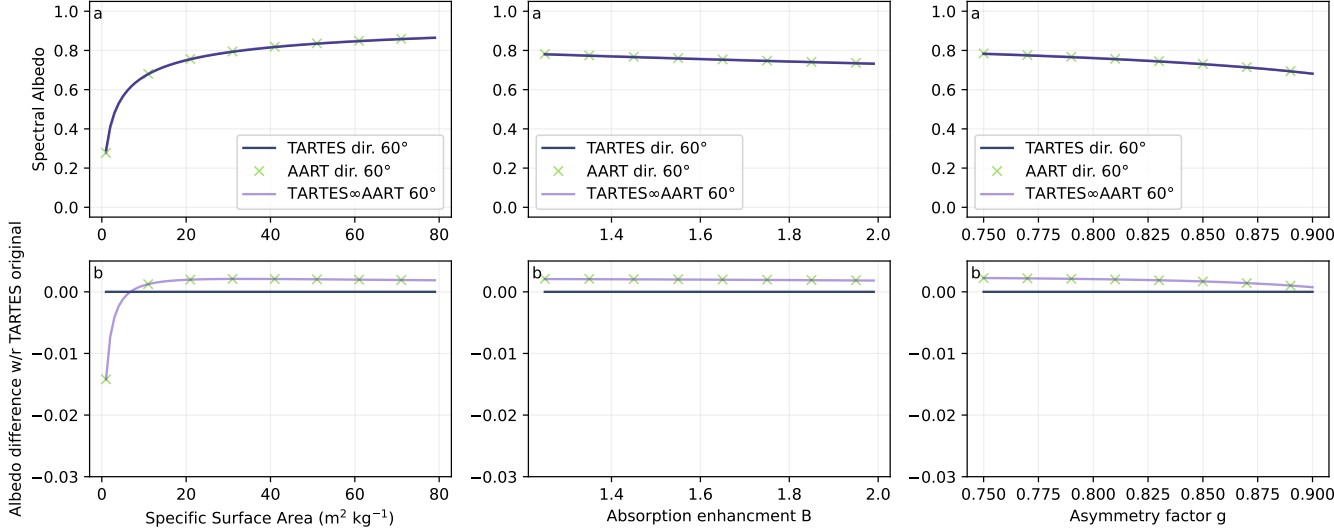

**Figure 7.** Comparison of direct albedo (SZA=60°) at $1300\,\mathrm{nm}$ as a function of SSA, $B_0$ and $g_0$ computed by TARTES and AART. The snowpack is semi-infinite with SSA = $20\,\mathrm{m^2\,kg^{-1}}$.

### 4.3 Comparison of TARTES with other numerical snow radiative transfer models

TARTES is now compared to two widely-used models, DISORT-Mie (with 16 streams) and SNICAR-ADv3 (online tool, https://snow.engin.umich.edu/, last access: 24 March 2024). Since the first model (and the second until a recent version) is limited to spherical particles, we consider this shape for all the simulations. In TARTES, this is achieved by letting $B_0 = 1.25$ and $g_0 = 0.895$ (Libois et al., 2013). TARTES.F is also included in this comparison.

### 4.4 Clean semi-infinite snowpack

Figure 8 shows diffuse albedo simulations for the different models for a semi-infinite snowpack with SSA = $20\,\mathrm{m^2\,kg^{-1}}$. Overall the agreement is very good, with virtually unnoticeable differences in Fig. 8a, except for TARTES∞AART that stands out for wavelengths higher than $1400\,\mathrm{nm}$. The residual albedo panel (Fig. 8b) reveals small differences of around 0.01 and occasionally up to 0.03 in amplitude, which may be significant for some applications. From this comparison no outliers nor particularly similar models emerge (except the new TARTES∞AART formulation). Furthermore the presence of spikes and 560 oscillations suggest numerical issues rather than physical differences or bugs. This is also suggested by the differences between the Python and Fortran versions of TARTES, despite a common theory and initial code. On average, TARTES appears closest to DISORT-Mie with a Root Mean Square Difference (RMSD) of 0.0035, then follow TARTES.F with a RMSD=0.0039 and SNICAR-ADv3 with a RMSD=0.0061. The agreement is overall much better in the shorter wavelengths < $1400\,\mathrm{nm}$ as also noted for the comparison with AART. The differences in broadband albedo reported in Fig. 8a are very small.





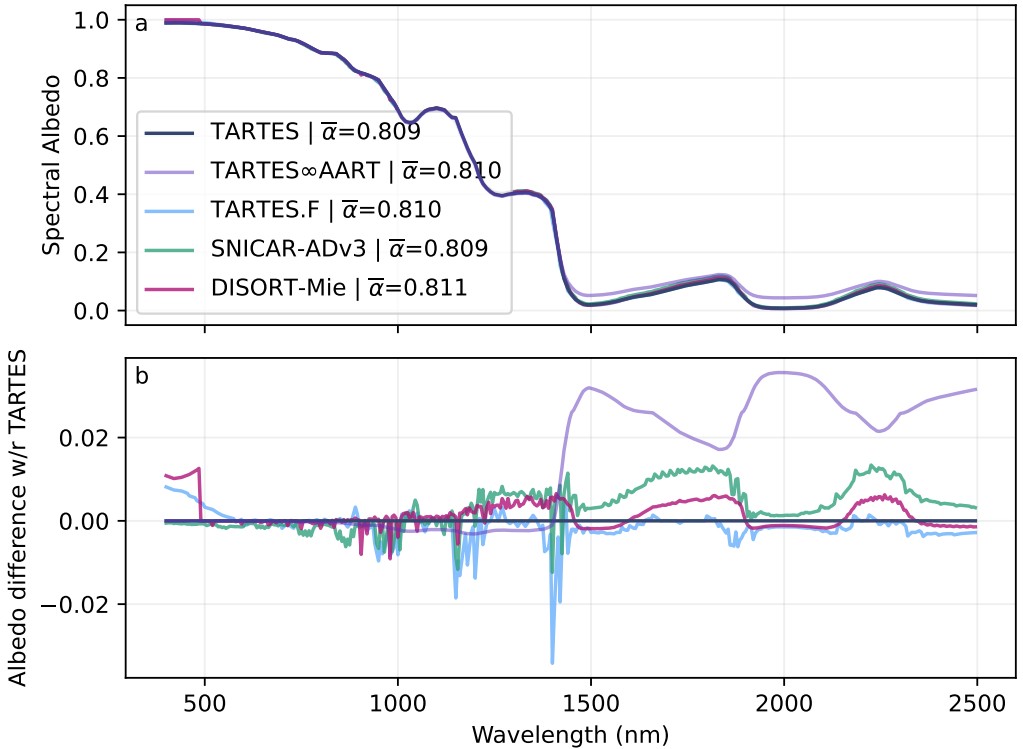

**Figure 8.** Comparison of diffuse albedo spectra calculated by different numerical models for a semi-infinite snowpack with SSA = 20 $\mathrm{m}^2 \, \mathrm{kg}^{-1}$.

For $\lambda < 490\,\mathrm{nm}$, DISORT-Mie yields an albedo of 1.0, which is certainly a rounding error. In this highly reflective domain, the 32-bit float arithmetic used by DISORT-Mie is certainly insufficient. The difference observed in the visible range for TARTES.F is explained by the use of the refractive index database (Warren and Brandt, 2008) (hard coded in the Fortran code) that has been recently updated with stronger absorption values (Picard et al., 2016), used for the other model simulations.

### 4.5   Clean two-layer snowpack

Figure 9 shows the comparison for a typical two-layer snowpack made of a thin layer of fresh snow (a 1 cm thick layer with SSA = $50\,\mathrm{m}^2\,\mathrm{kg}^{-1}$ and density $150\,\mathrm{kg}\,\mathrm{m}^{-3}$) on top of slightly aged snow (an infinitely-thick layer with SSA = $20\,\mathrm{m}^2\,\mathrm{kg}^{-1}$). SNICAR-ADv3 is not included because the online version used in this paper does not handle multiple layers. Overall the results are similar to the one layer case. The maximum difference is about 0.03, the average difference is much smaller, and the errors share some similar patterns with the former comparison.

As for the single-layered snowpack, TARTES∞AART stands out in the longer wavelengths. It was mainly implemented to check the conformity of TARTES with AART from a theoretical point of view, but appears to provide no practical benefice





over the original TARTES. For this reason, we do not further consider it. The original TARTES is kept as the default in the following, and in the code.

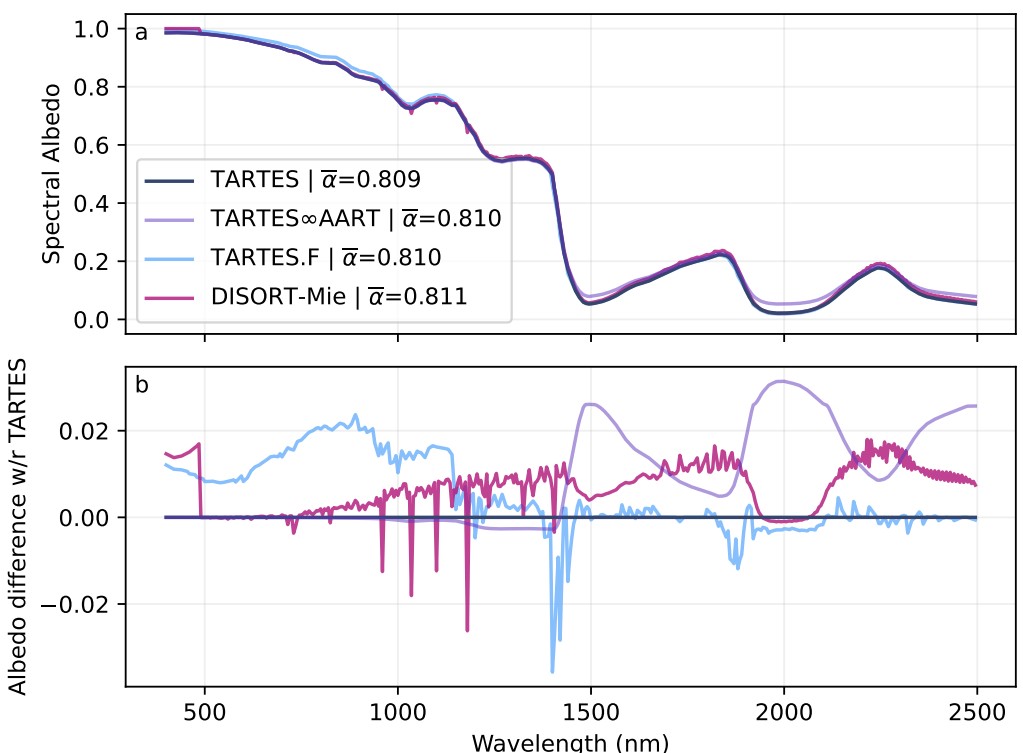

**Figure 9.** Comparison of diffuse albedo spectra calculated by different numerical models for a two-layer snowpack. A 1 cm layer with SSA = 50 m$^2$ kg$^{-1}$ and density=150 kg m$^{-3}$ is overlying a semi-infinite layer with SSA = 20 m$^2$ kg$^{-1}$.

### 4.5.1  Clean thin snowpack

The extreme case of a 1 cm thick snowpack with a perfectly black underlying surface (bottom albedo $\alpha_B$= 0) is presented in Fig. 10. In principle, the two-stream approximation is less adequate in these conditions (corresponding to $\tau^*$=7–17 depending on the wavelength) compared to the DISORT-Mie model. The results indeed show a degradation in the visible range, where the underlying surface has an impact (the albedo is notably lower than in Fig. 8). Nevertheless, the difference remains under 0.01 in this range, which is comparable to the more favorable case of the thick snowpack.

### 4.6  Snowpack polluted with black carbon and dust

Light absorbing impurities can be taken into account by all models considered here. Here we compare simulations with particles much smaller than the wavelength (BC) first, and then much larger than the wavelength (dust).



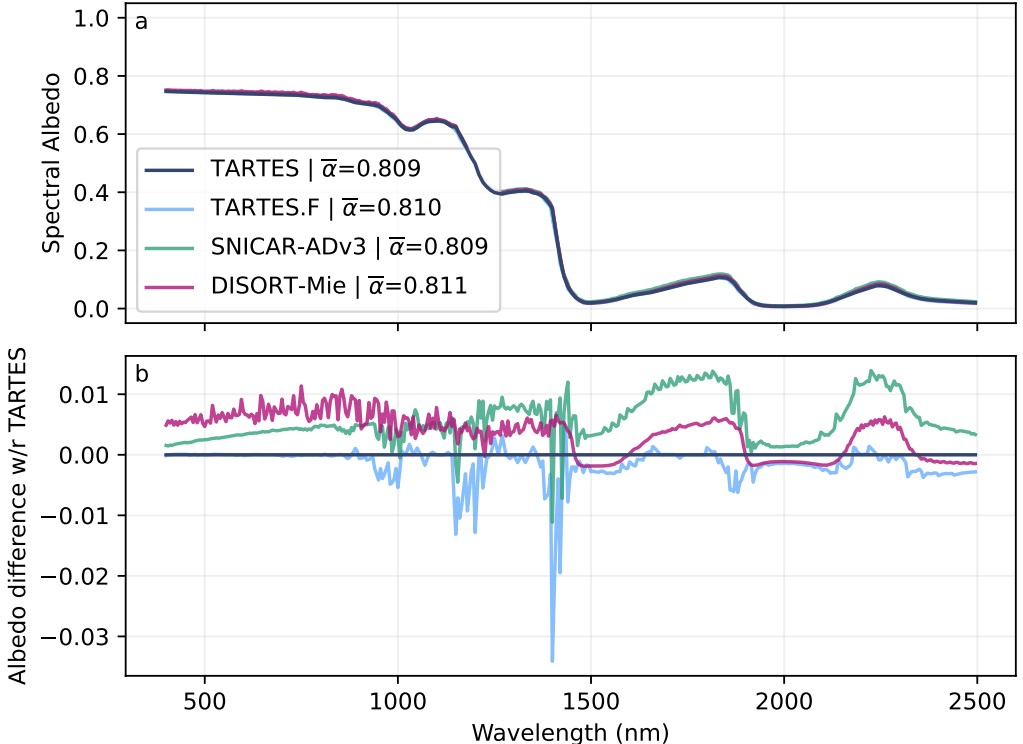

**Figure 10.** Comparison of diffuse albedo spectra calculated by different numerical models for a thin snowpack (1 cm thick) with SSA = 20 $\mathrm{m^2\,kg^{-1}}$ and density = $350\,\mathrm{kg\,m^{-3}}$ overlying a dark surface ($\alpha_b$ = 0).

Figure 11 shows semi-infinite diffuse albedo spectra obtained by the three models for BC concentrations of 100, 500 and 2000 $\mathrm{ng\,g^{-1}}$. For the sake of testing and validation, we use such extreme values compared to the typical amount found in snow (Bisiaux et al., 2012; Kang et al., 2020). Contrary to the clean snowpack, significant differences are obtained for concentrations of 500 and 2000 $\mathrm{ng\,g^{-1}}$. The difference reaches 0.04 around 500 nm between DISORT-Mie and TARTES at the maximal concentration. Such value, in the visible range where the solar irradiance is maximum, has a dramatic effect on the absorption. This is reflected in the differences of broadband albedo which are limited to 0.001-0.006 between SNICAR-ADv3 and TARTES but reach 0.005-0.022 between DISORT-Mie and TARTES depending on the concentration.

These differences are likely explained by the different representations of the impurities in these models. For maximal comparability, all the simulations assume the same refractive index and density of BC. They are taken from SNICAR-ADv3 (Flanner et al., 2021) which is also the default in TARTES v2.0. The three models also assume that the particles are suspended in the air (representation known as external mixing, Flanner et al., 2012). However, the size of the particles differs for reasons inherent to each model. SNICAR-ADv3 and DISORT-Mie uses Mie theory (spherical particles), however SNICAR-ADv3 simulates a log-normal distribution of particle sizes, with a mean mass-weighted radius of 67 µm for black carbon, while our implementa-




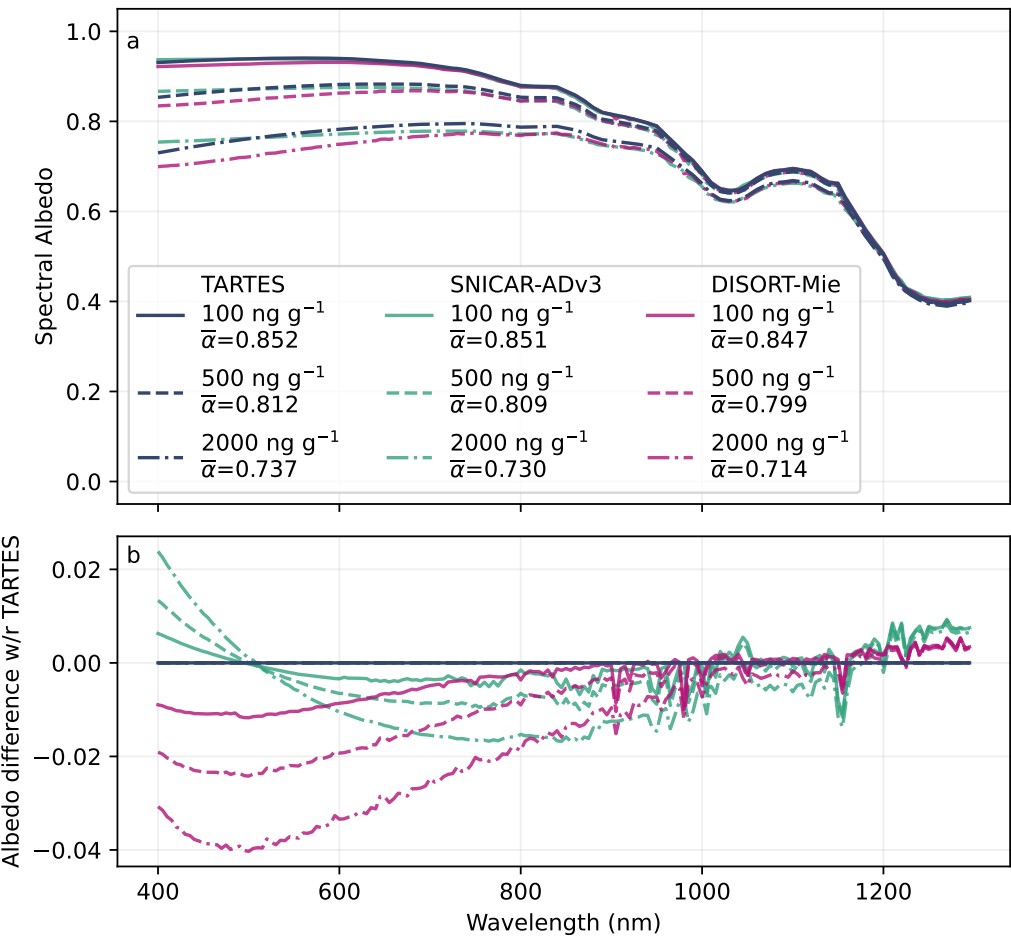

**Figure 11.** Comparison of diffuse albedo calculated by different numerical models for different BC concentration. The snowpack is semi-infinite, SSA = 20 m$^2$ kg$^{-1}$ and density = 350 kg m$^{-3}$.

tion of DISORT-Mie is limited to a mono-disperse collection of spheres. A radius of 67 μm is taken for Fig. 11 but this is not strictly equivalent to the SNICAR-ADv3 configuration. On the other hand, TARTES uses the Rayleigh approximation, that is only valid for small particles. To illustrate the influence of the particle size, Fig. 12 shows the variations of albedo at 550 nm predicted by DISORT-Mie. TARTES compares very well with DISORT-Mie for the extremely lowest radii (blue marker overlap the violet curve in Fig. 12), while SNICAR-ADv3 is slightly overestimating the DISORT-Mie albedo at 67 μm (green marker). Nevertheless, the variations of albedo in the range 0–150 nm are relatively small (< 0.02) , which seems acceptable with respect to the other uncertainties on the properties of the light absorbing particles (concentration and density).

Fig. 13 shows a comparison of different dust types (different origins and sizes) and models. The concentration is 100 μg g$^{-1}$, which is realistic in alpine snow (Dumont et al., 2020; Di Mauro et al., 2024). The agreement is again fairly good between





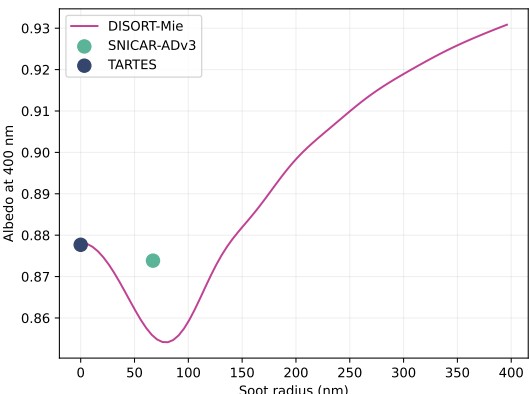

**Figure 12.** Diffuse albedo at $550\,\mathrm{nm}$ as a function BC particle radius, calculated by different numerical models. The snowpack is semi-infinite, BC concentration = $500\,\mathrm{ng\,g^{-1}}$, SSA = $20\,\mathrm{m^2\,kg^{-1}}$ and density = $350\,\mathrm{kg\,m^{-3}}$.

TARTES and SNICAR-ADv3 which results from the similar MAE values for the "Libya PM2.5" and "Algeria PM.25" dusts ($_{400\,\mathrm{nm}}$=110 and $73\,\mathrm{m^2 kg^{-1}}$) used in TARTES, compared to the values for Sahara used in SNICAR-ADv3 (Flanner et al., 2021, table 2, figure 3).

### 4.7 Simulations of profiles of absorption, irradiance and actinic flux.

Figure 14 illustrates TARTES ability to calculate profiles of absorption, irradiance and actinic flux for a three-layer snowpack

with SSA = $50, 20, 20\,\mathrm{m^2\,kg^{-1}}$ from top to bottom, density = $300, 300, 350\,\mathrm{kg\,m^{-3}}$ and thickness = $10, 20\,\mathrm{cm}$ with the last layer being infinitely thick. The wavelength is $600\,\mathrm{nm}$. The profiles are presented relative to the incident flux, this is why the x-axis label of each graph is unitless.

Regarding the absorption profile, the convention in TARTES is to return the total radiation absorbed in every layer which is suitable for a direct input in thermodynamic calculations. The unit is the same as that of the incident flux prescribed by the

user (variable *totflux*, usually $\mathrm{W\,m^{-2}\,nm^{-1}}$ in real applications). Regarding the actinic flux, the convention in TARTES is to return values in the same unit as the incident flux, which is prescribed by the user in the variable totflux. The conversion from spectral irradiance $\mathrm{W\,m^{-2}\,nm^{-1}}$ to actinic flux $\mathrm{photons\,s^{-1}\,cm^{-2}\,nm^{-1}}$ is let to the user. The irradiance and actinic flux profiles (Figs 14b, c) show a series of near linear decreasing trends in logarithm scale with varying slopes, which is equivalent to near exponential decreases in natural scale with varying rate. This rates is approximately the asymptotic extinction (Libois

et al., 2013) which can be deduced by approximating the AART extinction Eq 41 as:

$$k_e = \rho\sqrt{\frac{12B\chi\mathrm{SSA}(1-g^G)}{4\lambda\rho_{ice}}} \tag{84}$$

This formulation is implemented in the snowoptics package, function *extinction_KZ04*. Here we find $k_e =$9.14, 5.78, $8.09\,\mathrm{m^{-1}}$ for the layers from top to bottom, equivalent to e-folding depths ($l_e = 1/k_e$) of 11, 17 and $12\,\mathrm{cm}$ respectively. Calculating the





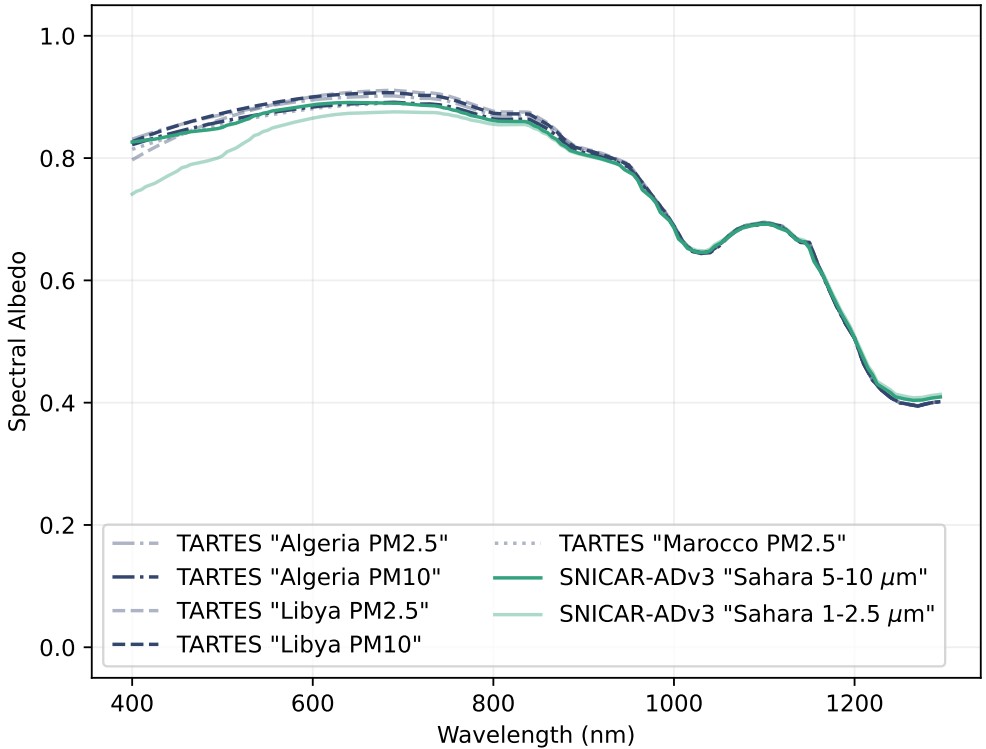

**Figure 13.** Comparison of diffuse albedo spectra calculated for different dust types and sizes, by two models. The snowpack is semi-infinite, dust concentration = $100\,\mu\mathrm{g\,g}^{-1}$ for all types, SSA = $20\,\mathrm{m}^2\,\mathrm{kg}^{-1}$ and density = $350\,\mathrm{kg\,m}^{-3}$.

vertical gradient of the irradiance logarithm from Fig. 14b (excluding $1\,\mathrm{cm}$ at the top and bottom of each layer) yields similar
values, $k_e = 11.1, 5.78, 8.09\,\mathrm{cm}$ respectively.

    The behavior of the irradiance gradient in the top layer is affected by the proximity of the surface (where the direct radiation is progressively converted into diffuse radiation) and the junction with the next layer that has different optical properties. This tends to bend the curve, meaning that the profile of irradiance is not exactly exponential. The actinic flux shows a similar behavior.

To conclude, this example illustrates that the profiles of irradiance and actinic flux can be approximated at first order by exponential decreases which decay can be calculated from snow properties (density, SSA, $B_0$ and $g_0$) in each layer (Eq. 84). However near the surface and in the presence of contrasted layers, it is recommended to use a proper multi-layered radiative transfer model as TARTES.



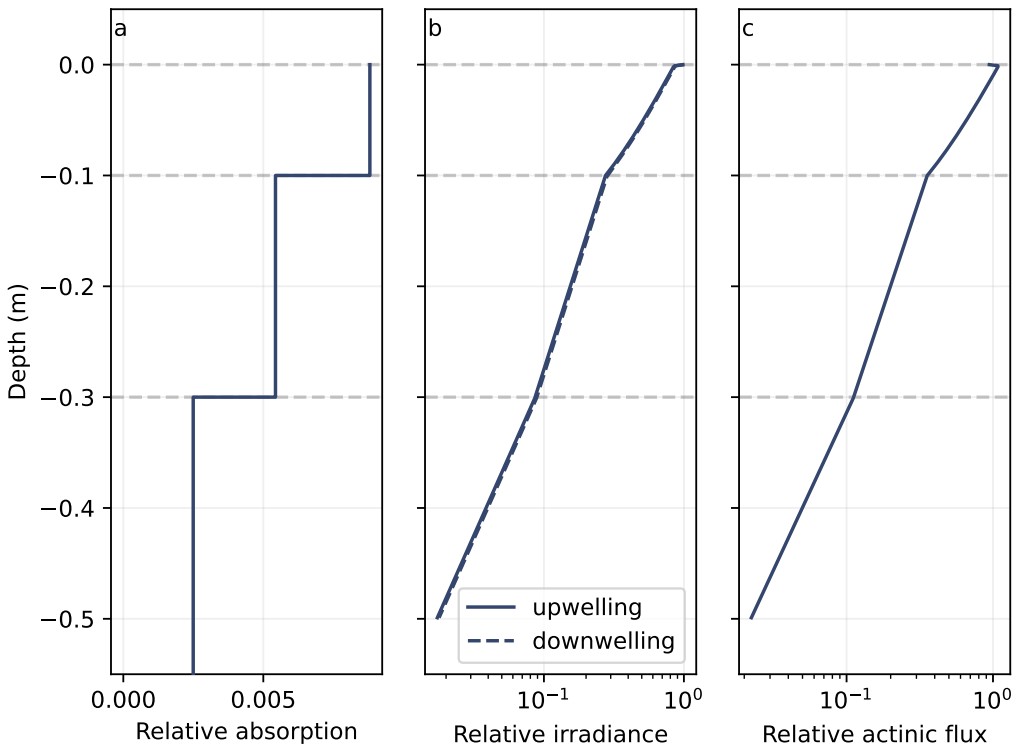

**Figure 14.** Profiles of a) absorption b) irradiance and c) actinic flux for a three-layer snowpack with SSA = 50, 20, 20 m$^2$ kg$^{-1}$, density = 250, 250, 350 kg m$^{-3}$ and thickness = 5 cm, 20 cm, $\infty$. The illumination is at 60° and the wavelength is 600 nm

## 5 Discussion

TARTES was designed to perform simple radiative transfer calculations in a plane parallel multi-layered snowpack, with the unique possibility to describe the shape of the particles using two parameters also used in the AART theory, namely $B$ the absorption enhancement parameter, and $g$ the asymmetry factor (Libois et al., 2013). This choice is important and motivated by two reasons. First these parameters are the main factors controlling the influence of the shape on the absorption and scattering properties in a weakly absorbing medium. Second, these parameters can be calculated for two-phase porous media, without

assuming that snow is a collection of particles with some given geometrical shape (Malinka, 2014; Robledano et al., 2023). While the physical meaning of $B$ and $g$ is often presented for single particles (e.g. Libois et al., 2013), the definition of these parameters is not tied to the notion of particle. For this reason we qualify these parameters as descriptive of the "optical shape of the snow" without requiring the medium to be actually composed of distinct particles (spheres, fractals, or cubes, ...). Moreover, it was found that $B = n^2$ applies very well to snow (Robledano et al., 2023) and only $g$ varies, but in a narrow range for snow

compared to across common geometrical shapes (spheres, fractals, hexagonal plates). Furthermore, the value of $B = n^2$ and the values of $g$ clearly indicate that snow does not behave as a collection of ice spheres. These results make TARTES inherently





more suitable to snow than Mie-based models. Note that most simulations presented in this paper used the values of $B$ and $g$ for spheres for the sole purpose of comparison with the established Mie-based models. In practice we do not recommend to run TARTES in these conditions. Instead we recommend $B = n^2$, and $g = 0.82$ which is the middle of the range found by Robledano et al. (2023). These values are the defaults in TARTES v2.0. Figure 15 illustrates the difference in albedo and irradiance at depth (20 cm) considering the default $B$ and $g$ values for snow, and for spheres. The albedo is higher for snow than for spheres for a given SSA and density, by 0.018 on average over the range 400–2000 nm, and reaches 0.042 at 1400 nm. These values are significant for surface energy budget calculations, with potent large impact in climate simulations (Räisänen et al., 2017). The irradiance at 20 cm depth is weaker for snow than for spheres, by about a factor 10 at 750 nm for instance. These differences can be explained by the strong forward scattering of spheres (high $g$) and the lower absorption enhancement parameter (low $B$) which tends overestimate the penetration at depth.

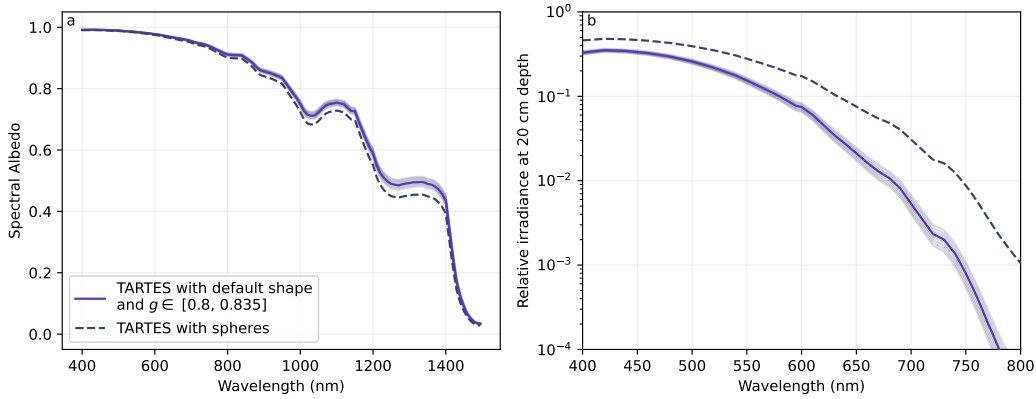

**Figure 15.** Spectral albedo a) and irradiance at 20 cm depth, relative to the incident irradiance at the surface b) for spheres, and snow (i.e. $B = n^2$ and $g \in [0.8, 0.835]$ according to Robledano et al. (2023)). The snowpack is semi-infinite with SSA = 20 m$^2$ kg$^{-1}$ and the illumination is at 60°.

Despite this advantage, TARTES presents some limitations owing to its simplicity. Consistent with the choice of only using the asymmetry factor $g$ instead of requiring the full phase function, the two-stream approximation was selected to solve the radiative transfer equation in TARTES instead of a multi-stream approach as in DISORT-based models. As a direct consequence, TARTES can not calculate the bi-directional reflectance (BRDF) which is essential for instance for satellite remote sensing applications. TARTES was initially designed for energy balance computations and only provides hemispherically averaged quantities, namely surface albedo, absorption in each layer, and profiles of upwelling and downwelling radiation flux and actinic flux.

As TARTES relies on the geometric approximation, the allowed range of ice particle sizes (or more generally, the length scales in the microstructure) is limited. The particles must be significantly larger than the wavelength (typically > 5 μm for the solar domain) which is usually valid for most snow types (Fierz et al., 2009; Walden et al., 2003). In contrast this assumption is invalid for light absorbing impurities such as BC, that usually come as sub-wavelength sized particles. It is still possible



to account for the absorption of these small particles, by neglecting their scattering. For very small particles, the Rayleigh approximation works well and allows a rigorous formulation of the absorption coefficient as a function of the particle complex refractive index and density. This approximation is acceptable for BC. For larger particles, as dust, TARTES relies on tabulated values of mass absorption efficiency (MAE) which can be obtained either from direct measurements or can be estimated by offline Mie calculations as in SNICAR. We believe that this simple treatment of light absorbing properties in snow is sufficient and of adequate complexity given the considerable uncertainties associated with the physical properties of light absorbing impurities and the difficulty to measure or simulate their concentration in snow.

Despite the differences between TARTES and the other models, the simulations of albedo presented in this paper (Sect. 4) generally show a good agreement, typically within 0.02. Notably, the errors between different models are typically lower than between the Python and FORTRAN versions of TARTES, which suggests that at this degree of agreement, most of the residual errors can result from implementation details and numerical issues rather than theoretical differences.

## 6  Conclusions

TARTES and the ecosystem of tools developed around this radiative transfer model for snow allow accurate simulations of several snow optical properties, most notably the spectral albedo and irradiance profiles in the snowpack. TARTES is intended to be user-friendly and easy to improve, thanks to the Python implementation. While technically the results of this paper demonstrate that TARTES performs equally well to other existing models when assuming snow as a collection of ice spheres, TARTES can handle a more general representation of snow that is more representative of natural snow than historical models based on idealized shapes to represent snow grains. For this particular reason, and despite the overall simplicity of the approximations implemented in TARTES, this model is able to accurately predict the optical properties of snow with given SSA and density, and is perfectly suited for implementation in atmospheric models, including climate models.

*Code availability.*  TARTES v2.0-pre0 used in this submitted manuscript is available from https://doi.org/10.5281/zenodo.12103855 and the latest development version is available from https://github.com/ghislainp/tartes (last access: 24 March 2024).

*Author contributions.*  QL wrote the theoretical formulation of TARTES. QL and GP developed TARTES Python version. GP ran the simulations and wrote the manuscript. Both authors commented the manuscript.

*Competing interests.*  the authors declare no competing interests



*Acknowledgements.* TARTES was initially developed as part of the Agence Nationale de la Recherche program MONISNOW no. 1-JS56-005-01-11JS56-005-01, and recent improvements were made possible through Agence Nationale de la Recherche program MiMESis-3D

project—grant no. ANR-19-CE01-0009). We thank Mathieu Lafaysse for providing the FORTRAN version that he implemented from the Python version of TARTES for integration in the open source model Crocus.



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
