# Peer review of "Simulation of snow albedo and solar irradiance profile with the two-stream radiative transfer in snow (TARTES) v2.0 model"

_EGUsphere, 2024_

## Author Comment (AC1)

Responses to Reviewer RC1

The authors provide a very comprehensive description of TARTES model, which serves as a very good technical documentation for the model. They also compared the TARTES model results with a few other widely-used snow radiative transfer models (AART, DISORT-Mie, SNICAR-ADv3), which generally show reasonable agreement (within 0.02 for snow albedo). Overall, the manuscript is well written. I have a few minor comments for the authors to address.

We thank the reviewer for the supportive comment. We have taken into account all the detailed corrections listed below, almost as proposed.

Minor comments:

1. It is not clear what the specific improvements in TARTES version 2 are compared to version 1, which needs to be clarified.

We propose to add a paragraph describing the scientific and the coding changes between version 2 and the previous version.

"Compared to the previous version, TARTES v2.0 proposes several options to compute diffuse radiation and to compute the semi-infinite layer albedo that appears in the two-stream equation. The default values for the grain shape parameters are changed based on recent advances (Robledano et al., 2023) and likewise for the black carbon properties to match SNICAR-ADv3.
The code has also been improved with the factorization of the impurities calculations (all types are now using MAE, either tabulated or calculated from the refractive index), type hinting, automatic strict formatting, modern packaging, and continuous integration for automatic publication on PyPI. The documentation has been improved and the conformity between the equations presented in this paper and the code carefully checked. In addition the SnowTARTES web application has been enhanced to calculate the irradiance profile."

2. It would be good to also summarize/discuss some underlying assumptions that may limit the TARTES model applications or the cautions users need to take when applying the model.

We propose to add more limitations in the dedicated paragraph of the discussion:

"Despite this advantage, TARTES presents some limitations owing to its simplicity. **It uses the conventional un-polarized radiative transfer, neglecting interferences, near-field and packing effects, as well as polarization effects. As a plane parallel model, the surface is supposed to be perfectly flat, surface roughness is neglected, which may impact simulations on rough terrain, especially at grazing angles. Likewise, the layers are perfectly smooth and horizontally semi-infinite. In practice, in areas where horizontal heterogeneity is strong, for instance as a result of snowdrift, this assumption might be inappropriate to simulate snow optical properties. Also, TARTES only considers snow excluding any other material that might be present in the snowpack, and models its optical behavior as a homogeneous scattering medium. It means that**

**the layers must be much thicker than the grain size (i.e. the free photon path). Regarding impurities, only their absorption is considered, and they are randomly distributed.** Consistent with the choice of only using the asymmetry factor g instead of requiring the full phase function, the two-stream approximation was selected to solve the radiative transfer equation in TARTES instead of a multi-stream approach as in DISORT-based models. As a direct consequence, TARTES can not calculate the bi-directional reflectance (BRDF) which is essential for instance for satellite remote sensing applications. TARTES was initially designed for energy balance computations and only provides hemispherically averaged quantities, namely surface albedo, absorption in each layer, and profiles of upwelling and downwelling radiation flux."

3. Lines 53-78: These widely-used snow albedo/radiative transfer models have been reviewed and described in details in He and Flanner (2020; https://doi.org/10.1007/978-3-030-38696-2_3), which would be a resource to mention and refer the interested readers to.

We propose to add the reference. Note however that this resource is under a paywall for us.

4. Equation (4): please introduce the delta function and F0 variable.

The definitions will be added: "where $F_0$ is the intensity of the solar beam at the surface and $\delta$ the Dirac function"

5. It would be good to have a table listing all the input and output variables (with units) for TARTES.

The table with symbols (used in the text), parameter names, description and unit will be added.

6. Equations (36-37): please give the mathematical expressions of "A" and "B".

They are unknowns to be determined from the boundary conditions later. We propose to add "where A and B are unknowns that will be determined according to the boundary conditions."

7. Line 244: "… we incorporate the exponential terms in the vector X". This sentence is not very clear to me.

The sentence is indeed not clear, we propose to remove it as the explicit expression for X, M and V are sufficient.

8. Equations (53-55): It seems that B is not included in these equations, which however should be, right?

The expression of X had indeed an error, it will be corrected by including B.

9. Line 300: For the negative albedo case, did the authors also reset other relevant quantities (e.g., internal light absorption energy, actinic flux, etc.)? If so, reset to what values?

We propose to add the following sentence to reply to this comment and stress that this option is only here for demonstrating a problem "The other quantities calculated by TARTES are not corrected, and we discourage to use the 2S option in general."

10. Line 320: Some studies (e.g., Peltoniemi, 2007: https://doi.org/10.1016/j.jqsrt.2007.05.009; He et al., 2017: https://doi.org/10.1002/2017GL072916) have quantified the impact of treating snow grains as densely packed medium on snow albedo, which is worth discussing briefly here.

To our understanding this topic is extremely complex as the effect of packing is many-fold. Different studies tend to address very different aspects (e.g. those cited by the reviewer and see also Doicu and Mishchenko. (2019) and Malinka (2023)), make different simplifying assumptions and draw different conclusions. How this applies to natural snow for practical applications is however difficult to assess.

Here, by considering that snow can be represented by the conventional radiative transfer equation applied to a homogeneous media (as done in TARTES, DISORT-Mie, SNICAR) we make a very strong assumption that automatically discards many of these packing effects. However, even within this simple framework, some dense packing effect on albedo has been reported under the geometrical optics approximation and perfectly random media (e.g. Malinka, A. (2023)) but the reason is still unclear to us.

Given the complexity of the topic and our lack of experience, we prefer not to address it at all. Instead we propose to make explicit that TARTES ignores the packing effect (see our response to comment 2).

References:
Doicu, A., & Mishchenko, M. I. (2019). An overview of methods for deriving the radiative transfer theory from the Maxwell equations. III: Effects of random rough boundaries and packing density. Journal of Quantitative Spectroscopy and Radiative Transfer, 224, 154–170. https://doi.org/10.1016/j.jqsrt.2018.11.002

Malinka, A. (2023). Stereological approach to radiative transfer in porous materials. Application to the optics of snow. Journal of Quantitative Spectroscopy and Radiative Transfer, 295, 108410. https://doi.org/10.1016/j.jqsrt.2022.108410

11. Section 3.2: It is not very clear what the key differences in model physics and/or parameters between the Python and Fortran versions, which needs to be clarified.

We propose to add: "TARTES.F was developed in 2014 exactly following the Python version. It contained the same physics and parameters. However it has not been updated since then, and now slightly differs from the most recent Python version. The main difference is the ice refractive database that is based on Warren et al. 2008 only (Picard et al. 2016 is not available). In 2019, impurities were added (Tuzet et al. 2017) using specific MAE values that slightly differ from the Python version (not used here)."

12. Line 506: It should note that this conclusion here is applied to semi-infinite snowpack tested in this study.

We also checked for a shallow snowpack (1 cm snowpack over a perfectly dark surface) and the same conclusion holds (see Figure below). We propose the following updated text:

"Based on these results **and calculations with a shallower snowpack (not shown)**, the direct 48.2° calculation was chosen as the default method to simulate diffuse radiation in TARTES. The integration is in principle more accurate but requires many more computations (solving the linear system for 128 angles instead of 1) even though measuring the execution time $(51\,\unit{ms}$ instead of $34\,\unit{ms})$ does not show a difference in the same proportion because only the constant vector of the linear system depends on the angle, not the matrix. In practice, users who prefer the accuracy offered by the integration method can explicitly set this option.

[Figure]

13. It would be good to add a short paragraph to discuss future plans for TARTES model improvements/developments.

We propose to add a list at the end of the Discussion:

"Possible future improvements in TARTES include the inclusion of terrain slopes, bubbly ice and slush layers, extended impurities database and internal mixture impurities."

---

## Author Comment (AC2)

Responses to Mark Flanner, reviewer RC2

This manuscript provides a comprehensive technical description of the TARTES v2.0 radiative transfer model for snowpack.  TARTES is widely used by the community and is embedded in the CROCUS snow thermodynamic model.  A unique and valuable feature of TARTES is the representation of ice particle asphericity via two parameters that can vary continuously and are not tied to any particular shapes, thus enabling the representation of the "optical shape" of snow via a continuum. The manuscript is well-written and appropriately includes both technical descriptions and comparisons against other snowpack radiative transfer models.  The introduction is well-referenced and provides useful background to the topic.  I have only minor comments and am happy to recommend publication of the manuscript in GMD.

We thank Mark Flanner for this main comment and the following comments, all of which we have thoroughly considered.

Minor comments:

line 30: "understand" -> "understanding"

done

lines 34-36: "SSA ... advantageously replaces the grain size as it can be rigorously defined and calculated for any porous medium" - While I agree about the advantages of SSA, "calculating" the surface area of complex shapes and porous media is often non-trivial.  To me, the main advantage of SSA is simply that it is a well-defined physical quantity, whereas the meaning of effective radius can be unclear for complex shapes and porous media.

Our intent in this sentence was to stress that SSA can be defined (and calculated) even when the medium is not made of grain (and so grain size is not well defined). We propose to amend the sentence by adding "even when distinct grains are not apparent".

line 107: "an" -> "a"

done

line 123: The latter part of the sentence also refers to diffuse illumination, so perhaps change the first party to "... illuminated by a beam source and diffuse light…"

done

line 163: "into" -> "in"

done

line 244: The appearance and application of these exponential terms is not immediately clear to me. Could you please clarify or elaborate on this point?

The reviewer RC1 made a similar comment, and we decided to remove this sentence as the equations are self-sufficient.

line 373: "When impurities are added in realistic low quantities, we assume the extinction coefficient of snow is unchanged..." - Although this is certainly a valid approximation for most realistic impurity mixing ratios, there may be situations where high impurity loads appreciably affect the total extinction coefficient, particularly in the near-IR spectrum. For example, very high dust loads can flatten out the 1.03um ice absorption feature (e.g., Fair et al, 2022, https://tc.copernicus.org/articles/16/3801/2022/) . Hence, it might be useful to provide a rough upper limit to the "low quantity" that applies for this assumption to hold, or in general to define the limits of applicability for model users.

It is difficult to provide rigorous limits without proper Mie-based numerical calculations, which TARTES does not do. Based on the reference suggested by the reviewer, showing results of such numerical calculations, we propose the following amendment:

"This supposes that impurity scattering is negligible. According to simulations at 1030\,\unit{nm} \citep{fair_2022}, this applies to BC in any case, as well as to dust except for fine particles (<1\unit{mu m}) in high concentration (e.g. >500\,\unit{ppm}). When this approximation is valid, it follows that the single scattering co-albedo is:"

line 466: Briefly, what are the user-controlled inputs to this "atmospheric_incident_spectrum" function?

There are two parameters:

"The function takes the solar zenith angle and cloud optical depth as input and uses the default cloud properties of SBDART \citep{ricchiazzi_1998}.

line 492: The accuracy of the delta-Eddington technique in handling diffuse incident light was also assessed more recently by Dang et al (2019, https://doi.org/10.5194/tc-13-2325-2019)

We propose to add the reference:

"\cite{wiscombe_1977} has shown that in the case of diffuse radiation the performances of the $\delta$-Eddington approximation were limited, sometimes leading to negative values of albedo. This is why \cite{warren_1980, **dang_2019**} computed the diffuse albedo as an angular average of direct albedos."

line 576: "benefice" -> "benefit"

done

p.29 and Fig. 11: The comparison provided is useful because it shows how the default representations of BC in each model affect the simulated albedo. Because the BC optical properties are slightly different in each model, one could further explore the sources of differences between TARTES and SNICAR by imposing identical BC properties. This could be accomplished, e.g., by directly importing the BC MAE values from the SNICAR optics library into TARTES, similar to how dust MAE is used in TARTES. I am not requesting this for manuscript revisions, but merely highlighting it as an informative sensitivity study for the future.

The MAE obtained with TARTES assumption (monodispersed, Rayleigh approximation) is close to SNICAR MAE reported in Fig 3a in Flanner et al. 2021 (see Figure below). The refractive index is exactly the same in both, so that the only difference is coming from the fact that SNICAR considers a distribution of size with Mie calculation, which leads to a slightly lower MAE in the short wavelength range (<500 nm), and higher in the longer range. This corresponds to the difference shown in our albedo comparison.

[Figure]

We propose the following addition in the text:

"Comparing MAE predicted by TARTES and reported for SNICAR-ADv3 in figure 3a in \citet{flanner_2021}, we observe (not shown) a slightly higher MAE with TARTES at 400\,\unit{nm} (+20\%), an agreement at 490\,\unit{nm}, and a slightly lower MAE at 1000\,\unit{nm} (-23\%), which explains the lower TARTES albedo at wavelengths <500\,\unit{nm} and higher at longer wavelengths observed in Fig. ~\ref{fig_model_comparison_soot}."

line 605: "67um": Typo, should be "67nm"

done

line 605: 67nm appears to be close to the monodisperse radius of maximum MAE. Because the SNICAR BC properties are mass-weighted from a lognormal distribution, the MAE value at the effective radius corresponding to the maximum MAE for a monodisperse distribution (i.e., the trough in Fig. 12) will always be less than the monodisperse MAE at that radius, since the weighted average incorporates lower MAE values from both sides of that radius. This could explain the phenomenon described in this part of the text. But other model factors could also contribute to differences in the absolute simulated albedos shown in Fig 12. For example, are identical ice refractive indices used in each model simulation?

We indeed use the same ice refractive index database for TARTES, SNICAR and DISORT. Only TARTES.F has its own database and was not updated for this work but it is not used here for the comparison with impurities.

We propose to add a more explicit statement about the use of the ice refractive index at the beginning of the section 4.3 on the comparison between models.

"The conditions of simulations are as similar as possible among the model. For instance the same ice refractive index \citep{picard_2016c} is used for the three models. TARTES.F is also included in this comparison, but with a different ice refractive index \citep{warren_2008} as discussed below."

We also propose to add an explanation why SNICAR gives a larger albedo than DISORT-Mie: "This results from the fact that SNICAR simulations with BC mean radius of 67 $\mu$m includes actual impurities with radii smaller and larger than 67 $\mu$m (it assumes a lognormal size distribution) for which the absorption is larger than at 67 $\mu$m, as 67 $\mu$m appears to be very close to the minimum absorption."

line 624: "This rates" -> "This rate"

done

line 638: "model as TARTES" -> "model such as TARTES"

done

line 654: For which wavelength or wavelength range does g=0.82 apply?

This applies to g0 at n=1.3 and any non absorbed wavelength (the input parameter of TARTES). In the discussion we decided not to distinguish g and g0 for simplicity.

line 655: "These values are the defaults in TARTES v2.0" - Actually, it might be helpful to include a table of all of the default parameter settings in TARTES v2.0, but I leave this to the authors to decide.

We added a table with the symbols in the revised paper, including the default values.

Finally, I agree with a comment from the other referee that it would be useful to highlight differences and improvements between versions 2.0 and 1.0 of TARTES.

We now detail the main differences between versions 1.0 and 2.0 of TARTES in section 3.1.

---

## Author Response (AR2)

Dear Editor,

We note that the reviewers do not have any comments at this stage.

However in this reponse, we would like to draw your attention on a bug which has been found by an external user of TARTES after the acceptance of the paper. This bug concerns the actinic flux code and only affects, marginally, figure 14 of the paper (see the difference below). The equations and the text are not affected at all. We propose to upload the new figure.

We would like to thank you and the reviewers for handling and reviewer our paper.

Best regards
Ghislain Picard & Quentin Libois

Reférence of the bug : https://github.com/ghislainp/tartes/issues/1#issuecomment-2413374422

[Figure]

Initial Fig 14                    New Fig 14